# Bootstrapped Representations in Reinforcement Learning

## Abstract

In reinforcement learning (RL), state representations are key to dealing with large or continuous state spaces. While one of the promises of deep learning algorithms is to automatically construct features well-tuned for the task they try to solve, such a representation might not emerge from end-to-end training of deep RL agents. To mitigate this issue, pretrained representations are often learnt from auxiliary tasks on offline datasets as part of an unsupervised pre-training phase to improve the sample efficiency of deep RL agents in a future online phase. Bootstrapping methods are today's method of choice to make these additional predictions but it is unclear which features are being learned. In this paper, we address this gap and provide a theoretical characterization of the pre-trained representation learnt by temporal difference learning (Sutton, 1988). Surprisingly, we find that this representation differs from the features learned by pre-training with Monte Carlo and residual gradient algorithms for most transition structures of the environment. We describe the goodness of these pre-trained representations to linearly predict the value function given any downstream reward function, and use our theoretical analysis to design new unsupervised pre-training rules. We complement our theoretical results with an empirical comparison of these pre-trained representations for different cumulant functions on the four-room (Sutton et al., 1999) and Mountain Car (Moore, 1990) domains and demonstrate that they speed up online learning.

## 1. Introduction

The process of representation learning is crucial to the success of reinforcement learning at scale. In deep rein-

[1]Anonymous Institution, Anonymous City, Anonymous Region, Anonymous Country. Correspondence to: Anonymous Author <anon.email@domain.com>.

Preliminary work. Under review by the International Conference on Machine Learning (ICML). Do not distribute.

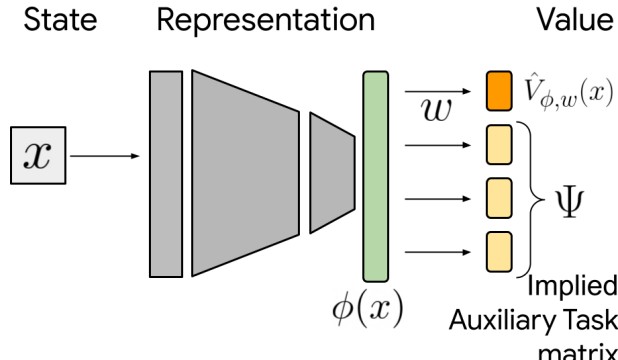

Figure 1: In deep RL, we see the penultimate layer of the network as the representation $\phi$ which is linearly transformed into a value prediction $\hat{V}_{\phi,w}$ and auxiliary predictions $\Psi(x)$ by bootstrapping methods.

forcement learning, a neural network is used to parameterise a representation $\phi$ which is linearly mapped into a *value function* (Figure 1) (Yu and Bertsekas, 2009; Bellemare et al., 2019; Levine et al., 2017); this approach often leads to state-of-the-art performance in the field (Mnih et al., 2015).

However, a representation supporting the downstream task of interest might not emerge from end-to-end training. Hence, auxiliary objectives are often incorporated into the training process to help the agent combine its inputs into useful features (Sutton et al., 2011; Jaderberg et al., 2017; Bellemare et al., 2017; Lyle et al., 2021) and the resulting network's representation can help the agent estimate the value function. Motivated by the desideratum of features rich enough for several downstream tasks, a number of works recently considered learning reward-agnostic representations from offline datasets of transitions with the environment (Stooke et al., 2021; Liu and Abbeel, 2021; Touati and Ollivier, 2021; Farebrother et al., 2023). The aim of this unsupervised pre-training step is that, when provided with a reward, the agent learns a control policy faster from these frozen representations than when trained in a tabula rasa way. While intuitively pretrained representations should capture temporal aspects of the environment, how prior computation should be leveraged in the form of frozen representations is lacking. This paper aims to fill this gap. We focus on unsupervised learning from auxiliary tasks consisting in predicting the expected return of a fixed policy

for several cumulant functions (Farebrother et al., 2023).

Because these tasks have mainly been trained by temporal difference learning (TD), we first analyze the features resulting from this kind of pretraining. In an idealized setting, we find that the way we train these value functions, for instance by TD learning, Monte Carlo or residual gradient, influences the resulting features (Section 3).

In Section 4, we quantify the approximation error of a linear prediction of the value function from these frozen representations in the TD learning and batch Monte Carlo settings. Given an unknown environment reward, we find that using an identity cumulant matrix theoretically leads to rich features. Because in practice we are interested in using a number of tasks much smaller than the number of states at the stage of unsupervised training, we provide an error bound that arises from sampling a small number of random pseudo-reward functions. In Subsection 5.3, we propose a novel unsupervised pretraining method with adapative cumulants and show that the resulting pretrained features outperform training from scratch on small scale domains.

## 2. Background

We consider a Markov decision process (MDP) $\mathcal{M} = \langle \mathcal{S}, \mathcal{A}, \mathcal{R}, \mathcal{P}, \gamma \rangle$ (Puterman, 1994) with finite state space $\mathcal{S}$, finite set of actions $\mathcal{A}$, transition kernel $\mathcal{P} : \mathcal{S} \times \mathcal{A} \to \mathscr{P}(\mathcal{S})$, deterministic reward function $\mathcal{R} : \mathcal{S} \times \mathcal{A} \to [-R_{\max}, R_{\max}]$, and discount factor $\gamma \in [0, 1)$. A stationary policy $\pi : \mathcal{S} \to \mathscr{P}(\mathcal{A})$ is a mapping from states to distributions over actions, describing a particular way of interacting with the environment. We denote the set of all policies by $\Pi$. We write $P_\pi : \mathcal{S} \to \mathscr{P}(\mathcal{S})$ the transition kernel induced by a policy $\pi \in \Pi$

$$P_\pi(s, s') = \sum_{s' \in \mathcal{S}} \mathcal{P}(s, a)(s')\pi(a \,|\, s)$$

and $r_\pi : \mathcal{S} \to [-R_{\max}, R_{\max}]$ the expected reward function

$$r_\pi(s) = \mathbb{E}_\pi[\mathcal{R}(S_0, A_0) \,|\, S_0 = s, A_0 \sim \pi(\cdot \,|\, S_0)].$$

For any policy $\pi \in \Pi$, the value function $V^\pi(s)$ measures the expected discounted sum of rewards received when starting from state $s \in \mathcal{S}$ and acting according to $\pi$:

$$V^\pi(s) := \mathbb{E}_{\pi, \mathcal{P}} \left[ \sum_{t=0}^\infty \gamma^t \mathcal{R}(S_t, A_t) \,|\, S_0 = s, A_t \sim \pi(\cdot \,|\, S_t) \right].$$

It satisfies the Bellman equation (Bellman, 1957)

$$V^\pi(s) = r_\pi(s) + \gamma \mathbb{E}_{S' \sim P_\pi(\cdot|s)}[V^\pi(S')],$$

which can be expressed using vector notation (Puterman, 1994) (viewing $r_\pi$ and $V^\pi$ as vectors in $\mathbb{R}^S$ and $P_\pi$ as an $\mathbb{R}^{S \times S}$ transition matrix) as

$$V^\pi = r_\pi + \gamma P_\pi V^\pi = (I - \gamma P_\pi)^{-1} r_\pi.$$

We are interested in approximating the value function $V^\pi$ using a linear combination of features (Yu and Bertsekas, 2009; Levine et al., 2017; Bellemare et al., 2019). We call the mapping $\phi : \mathcal{S} \to \mathbb{R}^d$ a *state representation*, where $d \in \mathbb{N}^+$. Usually, we are interested in reducing the number of parameters needed to approximate the value function and have $d \ll |\mathcal{S}|$. Given a feature vector $\phi(s)$ for a state $s \in \mathcal{S}$ and a weight vector $w \in \mathbb{R}^d$, the value function approximant at $s$ can be expressed as

$$V_{\phi,w}(s) = \phi(s)^\top w.$$

We write the *feature matrix* $\Phi \in \mathbb{R}^{S \times d}$ whose rows correspond to the per-state feature vectors $(\phi(s), s \in \mathcal{S})$. This leads to the more concise value function approximation

$$V_{\phi,w} = \Phi w.$$

In the classic linear function approximation literature, the feature map $\phi$ is held fixed, and the agent adapts only the weights $w$ to attempt to improve its predictions. By contrast, in deep reinforcement learning, $\phi$ itself is parameterized by a neural network and is typically updated alongside $w$ to improve predictions about the value function.

We measure the accuracy of the linear approximation $V_{\phi,w}$ in terms of the squared $\xi$-weighted $l_2$ norm, for $\xi \in \mathscr{P}(\mathcal{S})$,[1]

$$\|V_{\phi,w} - V^\pi\|_\xi^2 = \sum_{s \in \mathcal{S}} \xi(s)(\phi(s)^\top w - V^\pi(s))^2.$$

The $\xi$-weighted norm describes the importance of each state.

### 2.1. Auxiliary Tasks

In deep reinforcement learning, the agent can use its representation $\phi$ to make additional predictions on a set of $T$ auxiliary task functions $\{\psi_t \in \mathbb{R}^S\}_{t \in \{1, \dots, T\}}$ where each $\psi_t$ maps states to real values (Jaderberg et al., 2017; Bellemare et al., 2019; Dabney et al., 2021). These predictions are used to refine the representation itself. We collect these targets into an *auxiliary task matrix* $\Psi \in \mathbb{R}^{S \times T}$ whose rows are $\psi(s) = [\psi_1(s), \dots, \psi_T(s)]$. We are interested in the case of linear task approximation

$$\hat{\Psi} = \Phi W,$$

where $W \in \mathbb{R}^{d \times T}$ is a weight matrix, and want to choose $\Phi$ and $W$ such that $\hat{\Psi} \approx \Psi^\pi$. In this paper, we consider a variety of auxiliary tasks that ultimately involve predicting the value functions of auxiliary *cumulants*, also referred to as general value functions (GVFs; Sutton et al., 2011). By construction, these tasks can be decomposed into a non-zero cumulant function $g : \mathcal{S} \to \mathbb{R}^T$, mapping each state to

---

[1]We assume that $\xi(s) > 0$ for all states $s \in \mathcal{S}$.

$T$ real values, and an expected discounted next-state term when acting according to $\pi$

$$\psi^\pi(s) = g(s) + \gamma \mathbb{E}_{S' \sim P_\pi(\cdot|s)}[\psi^\pi(S')].$$

In matrix form, this recurrence can be expressed as follows

$$\Psi^\pi = G + \gamma P_\pi \Psi^\pi = (I - \gamma P^\pi)^{-1} G,$$

where $G \in \mathbb{R}^{S \times T}$ is a *cumulant matrix* whose columns correspond to each pseudo-reward vector. An example of a family of auxiliary tasks following this structure is the successor representation (SR) (Dayan, 1993). The SR encodes a state in terms of the expected discounted time spent in other states and satisfies the following recursive form

$$\psi^\pi(s, s'') = \mathbb{I}[s = s''] + \gamma \mathbb{E}_{S' \sim P_\pi(\cdot|s)}[\psi^\pi(S', s'')],$$

for all $s'' \in \mathcal{S}$. The SR is a collection of value functions associated with the cumulant matrix $G = I$. Here we focus our analysis in its tabular form, noting that it can be extended to larger state spaces in a number of ways (Barreto et al., 2017; Janner et al., 2020; Blier et al., 2021; Thakoor et al., 2022; Farebrother et al., 2023).

### 2.2. Monte Carlo Representations

To understand how auxiliary tasks shape representations, we start by presenting the simple case where the values of auxiliary cumulants are predicted in a supervised way. Here, the targets $\Psi^\pi = (I - \gamma P^\pi)^{-1} G$ are obtained by Monte Carlo rollouts, that is using the fixed policy to perform roll-outs and collecting the sum of rewards. The goal is to minimize the loss below

$$\mathcal{L}_{\text{aux}}^{\text{MC}}(\Phi, W) = \min_{W \in \mathbb{R}^{d \times T}} \|\Xi^{1/2}(\Phi W - \Psi^\pi)\|_F^2.$$

This method results in the network's representation $\Phi$, assuming a linear, fully-connected last layer, corresponding to the $k$ principal components of the auxiliary task matrix $\Psi^\pi$ if the network is other unconstrained (Bellemare et al., 2019).

**Proposition 1** (Monte Carlo representations). *If* rank$(\Psi^\pi) \geqslant d$, *all representations spanning the top-$d$ left singular vectors of $\Psi^\pi$ with respect to the inner product $\langle x, y \rangle_\Xi$ are global minimizers of $\mathcal{L}_{\text{aux}}^{\text{MC}}$ and can be recovered by stochastic gradient descent.*

In large environments, it is not practical to collect full trajectories to estimate $\Psi^\pi$. Instead, practitioners learn them by bootstrapping (Sutton and Barto, 1998).

### 2.3. Temporal Difference Learning with a Deep Network

Temporal difference (TD; Sutton, 1988) is the method of choice for these auxiliary predictions. The main idea of this approach is *bootstrapping* (Sutton and Barto, 1998). It consists in using the current estimate of the auxiliary task function to generate some targets replacing their true value $\Psi^\pi$ in order to learn a new approximant of the auxiliary task function. In this paper, we consider one-step temporal difference learning where we replace the targets by a one-step prediction from the currently approximated auxiliary task function. In deep reinforcement learning, both the representation $\phi$ and the weights $W$ are learnt simultaneously by minimizing the following loss function

$$\mathcal{L}_{\text{aux}}^{\text{TD}}(\phi, W) = \mathbb{E}_{\substack{s \sim \xi \\ s' \sim P_\pi(\cdot|s)}} \left[ \phi(s)W - \text{SG}(g(s) + \gamma\phi(s')W) \right]^2$$

where SG denotes a *stop gradient* and means that $\phi$ and $W$ are treated as a constant when taking the gradient from automatic differentiation tools (Bradbury et al., 2018; Abadi et al., 2016; Paszke et al., 2019). Written in matrix form, we have

$$\mathcal{L}_{\text{aux}}^{\text{TD}}(\Phi, W) = \|(\Xi)^{\frac{1}{2}}(\Phi W - \text{SG}(G + \gamma P^\pi \Phi W))\|_F^2$$

Here, $\Xi \in \mathbb{R}^{S \times S}$ is a diagonal matrix with elements $\{\xi(s) : s \in \mathcal{S}\}$ on the diagonal. For clarity of exposition, we express this loss with universal value functions but the analysis can be extented to state-action values at the cost of additional complexity. The idea is to reduce the mean squared error between the approximant $\hat{\psi}$ and the target values by stochastic gradient descent (SGD). Taking the gradient of $\mathcal{L}$ with respect to $\Phi$ and $W$, we obtain the *semi-gradient* update rule

$$\Phi \leftarrow \Phi - \alpha \Xi \left( (I - \gamma P^\pi)\Phi W - G \right) W^\top$$
$$W \leftarrow W - \alpha \Phi^\top \Xi \left( (I - \gamma P^\pi)\Phi W - G \right) \quad (1)$$

for a step size $\alpha$. Because the values of the targets change over time, the loss $\mathcal{L}$ does not have a proper gradient field (Dann et al., 2014) except in some particular cases (Barnard, 1993; Ollivier, 2018) and hence classic analysis of stochastic gradient descent (Bottou et al., 2018) does not apply.

## 3. Bootstrapped Representations

We now study the $d$-dimensional features that arise when performing value estimation of a fixed set of cumulants and how the choice of a learning method such as TD learning affects the learnt representations. Our first result characterizes representations that bootstrap themselves. We assume that the features $\Phi$ are updated in a tabular manner under the dynamics in Equation (1). To simplify the presentation, we now make the following invertibility assumption.

**Assumption 1.** *We assume that $\Phi^\top \Xi(I - \gamma P^\pi)\Phi$ is invertible for any full rank representation $\Phi \in \mathbb{R}^{S \times d}$.*

This standard assumption is for instance verified when $\xi$ is the stationary distribution over states under $\pi$ of an aperiodic, irreducible Markov chain (see e.g. Sutton et al., 2016).

Figure 2: **Left**: A simple 3-state MDP. **Right**: Five subspaces, each represented by a circle, spanned by $\Phi$ during the last training steps of gradient descent on $\mathcal{L}_{\text{aux}}^{\text{TD}}$ for $d = 2$.

An interesting characterization of the dynamical system in Equation (1) is its set of critical points. For a given $\Phi$, we write

$$W_{\Phi,G}^{\text{TD}} \in \{W \in \mathbb{R}^{d \times T} | \nabla_W \mathcal{L}_{\text{aux}}^{\text{TD}}(\Phi, W) = 0\}.$$

Using classic linear algebra, we find that the weights $W_{\text{TD}}$ obtained at convergence correspond to the LSTD solution (Bradtke and Barto, 1996; Boyan, 2002; Zhang et al., 2021)

$$W_{\Phi,G}^{\text{TD}} = \left(\Phi^{\mathsf{T}} \Xi (I - \gamma P^\pi) \Phi\right)^{-1} \Phi^{\mathsf{T}} \Xi G.$$

A key notion for our analysis is the concept of *invariant subspace* of a linear mapping.

**Definition 1** (Gohberg et al., 2006). *A representation $\Phi \in \mathbb{R}^{S \times d}$ spans a real invariant subspace of a linear mapping $M : S \to \mathbb{R}^{|S|}$ if the column span of $\Phi$ is preserved by $M$, that is in matrix form*

$$\text{span}(M\Phi) \subseteq \text{span}(\Phi).$$

For instance, any real eigenvector of $M$ generates one of its one-dimensional real invariant subspaces.

We are now equipped with the tools to enumerate the set of *critical representations* $\{\Phi \in \mathbb{R}^{S \times d} \,|\, \nabla_\Phi \mathcal{L}_{\text{aux}}^{\text{TD}}(\Phi, W_\Phi^{\text{TD}}) = 0\}$ in the lemma below.

**Lemma 1** (Critical representations for TD). *All full rank representations which are critical points to $\mathcal{L}_{\text{aux}}^{\text{TD}}$ span real invariant subspaces of $(I - \gamma P^\pi)^{-1} G G^{\mathsf{T}} \Xi$, that is* $\text{span}((I - \gamma P^\pi)^{-1} G G^{\mathsf{T}} \Xi \Phi) \subseteq \text{span}(\Phi).$

*Proof.* The proof is given in Appendix B and relies on the view of LSTD as an oblique projection (Scherrer, 2010). $\square$

In the particular case of an identity cumulant matrix and a uniform distribution over states, this set can be more directly expressed as the representations invariant under the transition dynamics.

**Corollary 1.** *If $G = I$ and $\Xi = I/|S|$, all full rank representations which are critical points to $\mathcal{L}_{\text{aux}}^{\text{TD}}$ span real invariant subspaces of the invariant subspaces of $P^\pi$.*

Similarly to how the top principal components of a matrix explain most of its variability (Hotelling, 1933), these critical representations are not equally informative of the dynamics of the environment. This motivates the need to understand the behavior of the updates from Equation (1). To ease the analysis, we assume that the weights $W$ have converged perfectly to $W_{\Phi,G}^{\text{TD}}$ at each time step (Le Lan et al., 2022a) and consider the following continuous-time dynamics.

$$\frac{d}{dt}\Phi = -\nabla_\Phi \mathcal{L}(\Phi, W_{\Phi,G}^{\text{TD}}) = -F(\Phi), \qquad (2)$$

where:

$$F(\Phi) := 2\Xi \left((I - \gamma P^\pi)\Phi W_{\Phi,G}^{\text{TD}} - G\right)(W_{\Phi,G}^{\text{TD}})^{\mathsf{T}}.$$

Our key result is that the stable critical points of this ordinary differential equation correspond to the *real* top-$d$ invariant subspace of $P^\pi$, when this exists.

**Theorem 1** (TD representations). *Assume $G = I$ and a uniform distribution $\xi$ over states. Let $\lambda_1, .., \lambda_{|S|}$ be the (possibly complex) eigenvalues of $P^\pi$, ordered by decreasing real part $\text{Re}(\lambda_i) \geqslant \text{Re}(\lambda_{i+1})$, $i \in \{1, .., |\mathcal{S}|\}$. Under the dynamics in Equation (2), all real invariant subspaces of dimension $d$ are critical points, and only the top-$d$ invariant subspace, if it exists, is stable.*

The result above implies that the TD algorithm converges towards a real top-$d$ invariant subspace or diverges with probability 1. While real diagonalisable transition matrices always induce real invariant subspaces, complex eigenvalues do not guarantee their existence and in such a case, where there is no top-$d$ real invariant subspace, the representation learning algorithm *does not converge*. As an illustration, consider the three-state MDP depicted in Figure 2, left, whose transition matrix is complex diagonalisable and given by

$$P^\pi = \begin{bmatrix} 0 & 1 & 0 \\ 0 & 0 & 1 \\ 1 & 0 & 0 \end{bmatrix}$$

Its eigenvalues are $\lambda_1 = 1$ associated to the real eigenvector $e_1$ and the complex conjugate pair $(\lambda_2, \overline{\lambda_2}) = (e^{2\pi i/3}, e^{-2\pi i/3})$, associated to the pair of real eigenvectors $(e_2, e_3)$. Hence, the real invariant subspaces of $P^\pi$ are $\{0\}, \text{span}(e_1), \text{span}(e_2, e_3), \text{span}(e_1, e_2, e_3)$. Note that there is no 2-dimensional real invariant subspace containing the top eigenvector $e_1$. Consequently, the 2-dimensional representation learnt by gradient descent on the TD learning rule with $G = I$ does not converge and rotates in the higher dimensional subspace $\text{span}(e_1, e_2, e_3)$ (see Figure 2, right).

| MAIN ALGORITHM | $l_1$-BALL OPTIMAL REPRESENTATION | REPRESENTATION LOSS | LEARNT REPRESENTATION |
|---|---|---|---|
| BATCH MC | SVD $\left((I - \gamma P^\pi)^{-1}\right)$ | MC | SVD $\left((I - \gamma P^\pi)^{-1} G\right)$ |
| RESIDUAL | SVD $\left((I - \gamma P^\pi)^{-1}\right) \Sigma_d$ | RESIDUAL | $(I - \gamma P^\pi)^{-1}$ SVD $(G)$ |
| TD | $\Phi_{\text{TD}}^*$ | TD | INV $\left((I - \gamma P^\pi)^{-1} G G^\mathsf{T} \xi\right)$ |

Table 1: Different types of representation loss and their induced representations. The supervised targets $\Psi \in \mathbb{R}^{S \times T}$ are $(I - \gamma P^\pi)^{-1} G$. SVD(M) denotes the top-$d$ left singular vectors of M, INV(M) the top-$d$ invariant subspace of M and $\Sigma_d \in \mathbb{R}^{d \times d}$ the diagonal matrix with the top-$d$ singular values of $(I - \gamma P^\pi)^{-1}$ on its diagonal.

To understand the importance of the stop-gradient in TD learning, it useful to study the representations arising from the minimization of the following loss function

$$\mathcal{L}_{\text{aux}}^{\text{res}}(\Phi, W) = \left\| \Xi^{\frac{1}{2}} \left(\Phi W - (G + \gamma P^\pi \Phi W)\right) \right\|_F^2,$$

which corresponds to residual gradient algorithms (Baird, 1995). While it has been remarked on before that the weights minimizing $\mathcal{L}_{\text{aux}}^{\text{res}}(\Phi, W)$ for a fixed $\Phi$ differ from $W_{\Phi, G}^{\text{TD}}$ (see Lemma 8; Lagoudakis and Parr, 2003; Scherrer, 2010), this objective function also has a different optimal representation

**Proposition 2** (Residual representations). *Let $d \in \{1, ..., S\}$ and $F_d$ be the top $d$ left singular vectors of $G$ with respect to the inner product $\langle x, y \rangle_\Xi = y^\mathsf{T} \Xi x$, for all $x, y \in \mathbb{R}^{|\mathcal{S}|}$. All representations spanning $(I - \gamma P^\pi)^{-1} F_d$ are global minimizers of $\mathcal{L}_{\text{aux}}^{\text{res}}$ and can be recovered by stochastic gradient descent.*

While TD and Monte Carlo representations are in general different, in the particular case of symmetric transition matrices and orthogonal cumulant matrices, they are the same.

**Corollary 2** (Symmetric transition matrices). *If a cumulant matrix $G \in \mathbb{R}^{S \times T}$ (with $T \geqslant S$) has unit-norm, orthogonal columns (e.g. $G = I$), the representations learnt from the supervised objective $\mathcal{L}_{\text{aux}}^{\text{MC}}$ and the TD update rule $\mathcal{L}_{\text{aux}}^{\text{TD}}$ are the same for symmetric transition matrices $P^\pi$ under a uniform state distribution $\xi$.*

This is because eigenvectors and singular vectors are identical in that setting and the eigenvalues of the successor representation are all positive.

## 4. Representations for Policy Evaluation

With the results from the previous section, the question that naturally arises is which approach results in better representations. To provide an answer, we consider a two-stage procedure. First, we learn a representation $\Phi$ by predicting the values of $T$ auxiliary cumulants simultaneously, using one of the learning rules described in Section 3. Then, we retain this representation and perform policy evaluation. If the value function is estimated on-policy, it converges towards

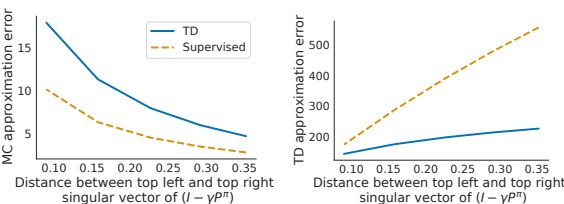

Figure 3: MC (**left**) and TD (**right**) approximation errors as a function of the misalignment of the top left and right singular vector of the SR induced by greedifying the policy. Trained with $\mathcal{L}_{\text{aux}}^{\text{MC}}$, $\mathcal{L}_{\text{aux}}^{\text{TD}}$, $G = I$, $d = 1$ on a 4-state room.

the LSTD solution (Tsitsiklis and Van Roy, 1996)

$$\hat{V}^{\text{TD}} = \Phi w_\Phi^{\text{TD}}$$

where $w_\Phi^{\text{TD}} = \left(\Phi^\mathsf{T} \Xi (I - \gamma P^\pi) \Phi\right)^{-1} \Phi^\mathsf{T} \Xi r_\pi$. We are interested in whether this value function results in low approximation error on average over random reward functions $r_\pi$, that is we want the following error to be small

$$\mathbb{E}_{r_\pi}[\|\Phi w_\Phi^{\text{TD}} - V^\pi\|_\xi^2] \tag{3}$$

where the expectation is over the reward functions $r_\pi$ sampled uniformly over the $l_1$ ball $\{r_\pi \in \mathbb{R}^S \mid \|r_\pi\|_1 \leqslant 1\}$. This set models an unknown reward function.

We say that a representation $\Phi_{\text{TD}}^*$ is $l_1$-*ball optimal* for TD learning when it minimizes the error in Equation (3). Here $\Phi_{\text{TD}}^*$ depends on the transition dynamics of the environment but not on the reward function.

**Lemma 2.** *A representation $\Phi_{\text{TD}}^*$ is $l_1$-ball optimal for TD learning iff it is a solution of the following optimization problem.*

$$\Phi_{\text{TD}}^* \in \arg\min_\Phi \left\| \Xi^{1/2} (\Phi W_{\Phi, I}^{\text{TD}} - (I - \gamma P^\pi)^{-1}) \right\|_F^2.$$

When $P^\pi$ is symmetric and $\Xi = I/|\mathcal{S}|$, the minimum is achieved by both the top-d left singular vectors and top-$d$ invariant subspace of the SR. However, as the misalignment between the top-$d$ left and top-$d$ right singular vectors of $(I - \gamma P^\pi)$ increases, the top-$d$ invariant subspace results in lower error compared to the top-$d$ singular vectors (see

Figure 3); note that here, none of them achieves $\Phi^*_{\text{TD}}$ and hence $G = I$ is not optimal for TD learning.

As a comparison, we study which representations are $l_1$-ball optimal for linear batch Monte Carlo policy evaluation. In that setting, we are given a dataset consisting of states and their associated value, which can be estimated by the realisation of the random return (Bellemare et al., 2017; Sutton and Barto, 2018), and the weights are learnt by least square regression. As above, we want the features minimizing

$$\mathbb{E}_{r_\pi}[\|\Phi w^{\text{MC}}_\Phi - V^\pi\|^2_\xi] \qquad (4)$$

where $\hat{V}^{\text{MC}} = \Phi w^{\text{MC}}_\Phi$ is the value function learnt at convergence and $w^{\text{MC}}_\Phi = (\Phi^\top \Xi \Phi)^{-1}\Phi^\top \Xi V^\pi$.

**Lemma 3.** *A representation $\Phi^*_{\text{MC}}$ is $l_1$-ball optimal for batch Monte Carlo policy evaluation if its column space spans the top-$d$ left singular vectors (with respect to the inner product $\langle x, y\rangle_\Xi$) of $(I - \gamma P^\pi)^{-1}$.*

Unlike TD, $\Phi^*_{\text{MC}}$ is achieved by training $\mathcal{L}^{\text{MC}}_{\text{aux}}$ with $G = I$.

We summarize in Table 1 our representation learning results mentioned throughout Section 3 and Section 4. For completeness, we also include $l_1$-ball optimal representations for residual algorithms. Proofs can be found in Appendix C.

### 4.1. TD and Monte Carlo Need Different Cumulants

Having characterized which features common auxiliary tasks capture and what representations are desirable to support training the main value function, we now show that MC policy evaluation and TD learning need different cumulants. In large environments, we are interested in cumulant matrices encoding a small number of tasks $T \ll S$.

**Lemma 4.** *Denote $B_T$ the top-$T$ right singular vectors of the SR and $\mathcal{O}(T, S)$ the set of orthogonal matrices in $\mathbb{R}^{T \times S}$. Training auxiliary tasks in a MC way with any $G$ from the set $\{G \in \mathbb{R}^{S \times T} | \exists M \in \mathcal{O}(T, S), G = B_T M\}$ results in an $l_1$-ball optimal representation for batch Monte Carlo.*

We showed in Section 3 that training auxiliary tasks by TD does not always converge when the transition matrix has complex eigenvalues. Maybe surprisingly, we find that this is not problematic when learning the main value function by TD. Indeed, the rotation of its own weights balances the rotation of the underlying representation.

**Lemma 5.** *Let $\{\Phi_\omega\}$ be the set of rotating representations from Figure 2 learnt by TD learning with $G = I$ and $d = 2$. All these representations are equally good for learning the main value function by TD learning, that is $\forall \omega \in [0, 1]$,*

$$\mathbb{E}_{\|r\|^2_2 < 1}\left\|\Phi_\omega w^{\text{TD}}_{\Phi_\omega} - V^\pi\right\|^2_F$$

*is constant and independent of $\omega$.*

Although $G = I$ does not always lead to $\Phi^*_{\text{TD}}$ when training $\mathcal{L}^{\text{TD}}_{\text{aux}}$, by analogy with the MC setting, we assume that $G = I$ leads to overall desirable representations. Assuming $\Xi = I/|\mathcal{S}|$, this means we would like the subspace spanned by top-$d$ invariant subspaces of $(I - \gamma P^\pi)^{-1}$ to be the same as the subspace spanned by the top $d$ invariant subspaces of $(I - \gamma P^\pi)^{-1}GG^\top$.

**Lemma 6.** *The set of cumulant matrices $G \in \mathbb{R}^{S \times T}$ that preserve the top-$T$ invariant subspaces of the successor representation by TD learning are the top-$T$ orthogonal invariant subspaces of $(I - \gamma P^\pi)^{-1}$, that is satisfying $G^\top G = I$ by orthogonality and $(I - \gamma P^\pi)^{-1}G \subseteq G$ by the invariance property.*

Unlike the MC case, a desirable cumulant matrix should encode the exact same information as the representation being learnt and the choice of a parametrization here matters.

### 4.2. A Deeper Analysis of Random Cumulants

We now study random cumulants which have mainly been used in the literature (Dabney et al., 2021; Lyle et al., 2021; Farebrother et al., 2023) as a heuristic to learn representations. We aim to explain their recent achievements as a pretraining technique (Farebrother et al., 2023) and their effectiveness in sparse reward environments (Lyle et al., 2021).

**Proposition 3** (MC Error bound)**.** *Let $G \in \mathbb{R}^{S \times T}$ be a sample from a standard gaussian distribution and assume $d \leqslant T$. Let $F_d$ be the top-$d$ left singular vectors of the successor representation $(I - \gamma P^\pi)^{-1}$ and $\hat{F}_d$ be the top left singular vectors of $(I - \gamma P^\pi)^{-1}G$. Denote $\sigma_1 \geqslant \sigma_2 \geqslant ... \geqslant \sigma_S$ the singular values of the SR and $\text{dist}(F_d, \hat{F}_d)$ the $\sin \theta$ distance between the subspaces spanned by $F_d$ and $\hat{F}_d$. We have*

$$\mathbb{E}[\text{dist}(F_d, \hat{F}_d)] \leqslant \sqrt{\frac{d}{T - d - 1}}\frac{\sigma_{d+1}}{\sigma_d} + \frac{e\sqrt{T}}{T - d}\left(\sum_{j=d+1}^{n}\frac{\sigma_j^2}{\sigma_d^2}\right)^{\frac{1}{2}}$$

*Proof.* A proof can be found in Appendix D and follows arguments from random matrix theory. $\square$

This bound fundamentally depends on the ratio of the singular values $\sigma_{d+1}/\sigma_d$ of the successor representation. As the oversampling parameter $(T - d)$ grows, the right hand side tends towards 0. In particular, for the right hand side to be less than $\epsilon$, we need the oversampling parameter to satisfy $(T - d) \geqslant 1/\epsilon^2$. We investigate to which extent this result holds empirically for the TD objective in Subsection 4.1.

## 5. Empirical Analysis

In this section, we illustrate empirically the correctness of our theoretical characterizations from Section 3 and compare the goodness of different cumulants on the four room

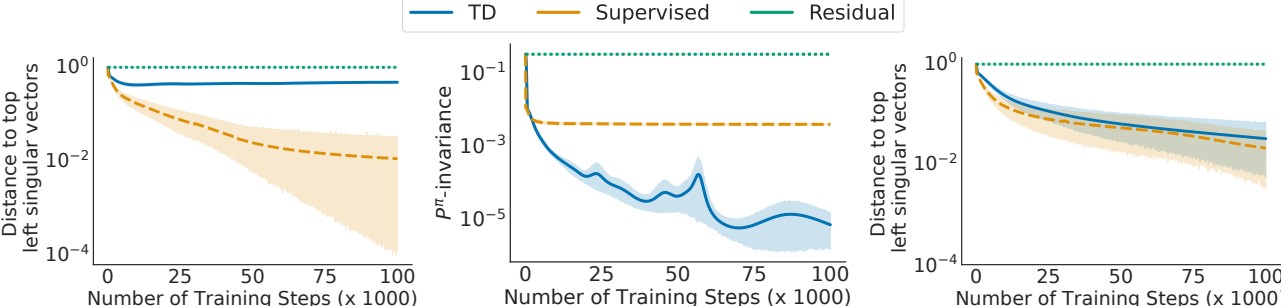

Figure 4: Subspace distance between $\Phi$ and the top-$d$ left singular vectors of the SR on the **left** (resp. and a top-$d$ $P^\pi$-invariant subspace in the **middle** over the course of training $\mathcal{L}_{\text{aux}}^{\text{TD}}$, $\mathcal{L}_{\text{aux}}^{\text{MC}}$ and $\mathcal{L}_{\text{aux}}^{\text{res}}$ for $10^5$ steps, averaged over 30 seeds ($d = 3$). MDPs with real diagonalisable (**left, middle**) and symmetric (**right**) transition matrices are randomly generated. Shaded areas represent $95\%$ confidence intervals.

(Sutton and Barto, 2018) and Mountain car (Moore, 1990) domains. Let $P_\Phi = \Phi(\Phi^\top \Phi)^\dagger \Phi^\top$. Here, any distance between two subspaces $\Phi$ and $\Phi^*$ is measured using the normalized subspace distance, [2] (Tang, 2019) defined by

$$\text{dist}(\Phi, \Phi^*) = 1 - \frac{1}{d} \cdot \text{Tr}\left(P_{\Phi^*} P_\Phi\right) \in [0, 1].$$

### 5.1. Synthetic Matrices

To begin, we train the TD, supervised and residual update rules from Section 3 up to convergence knowing the exact transition matrices $P^\pi$. In Figure 4 left and middle, we randomly sample 30 real diagonalisable matrices $P^\pi \in \mathbb{R}^{50 \times 50}$ to prevent any convergence issue from the TD update rule. In Figure 4 right, we generate symmetric transition matrices $P^\pi \in \mathbb{R}^{50 \times 50}$. To illustrate the theory, we run gradient descent on each learning rule by expressing the weights implicitly as a function of the features (see Equation (2) for TD for instance). Figure 4, left shows that these auxiliary task algorithms learn different representations and successfully recover our theoretical characterizations from Table 1. Figure 4 right illustrates that the supervised and TD rules converge to the same representation for symmetric $P^\pi$, as predicted by our theory.

### 5.2. Efficacy of Random Cumulants

Following our theoretical analysis from Subsection 5.2, our aim is to illustrate the goodness of random cumulants at recovering the left singular vectors of the successor representation on the four room domain (Sutton et al., 1999) and to investigate to which extent an analogous result holds empirically for the TD rule. We investigate the importance of three properties of a distribution: isotropy, norm and orthogonality of the columns. We consider random cumulants from different distributions: a standard Gaussian $\mathbb{N}(0, I)$, a Gaussian distribution which columns are normalized to

---

[2]It is equivalent to the $\sin \theta$ distance up to some constant

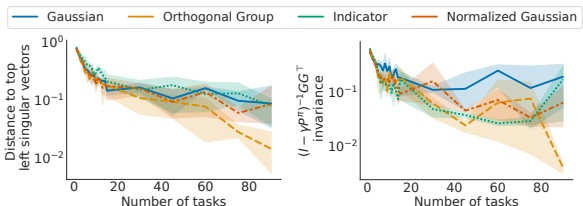

Figure 5: Subspace distance after $5 \times 10^5$ training steps ($d = 5$) between $\Phi$ learnt with $\mathcal{L}_{\text{aux}}^{\text{MC}}$ and the top left singular vectors of the SR (**left**) and between $\Phi$ learnt with $\mathcal{L}_{\text{aux}}^{\text{TD}}$ and the top invariant subspaces of the SR (**right**) for different random cumulants, on the four-room domain. Shaded areas represent $95\%$ confidence intervals

be unit-norm, the $O(N)$ Haar distribution and random indicators functions. Figure 5, left shows that the the indicator distribution which is not isotropic performs worse overall for the supervised objective and when the number of tasks is large enough, orthogonality between the columns of the cumulant matrix leads to better accuracy. In comparison, Figure 5, right studies the goodness of random cumulants at recovering the top-$d$ invariant subspaces of the SR and depicts a different picture. Here, the Gaussian distribution achieves the highest error irrespective of the number of tasks sampled while the normalized Gaussian achieves lower error suggesting the norm of the columns matter for TD training. The indicator distribution performs well for many number of sampled tasks indicating that the isotropy of the distribution is not as important for TD as it is for supervised training. Finally, the orthogonal cumulants achieve the lowest error when the number of tasks is large enough, showing this is an important property for both kinds of training.

### 5.3. Offline Pre-training

In this section we follow a similar evaluation protocol as that of Farebrother et al. (2023), but applied to the four room and Mountain car domains to allow a clear investigation of

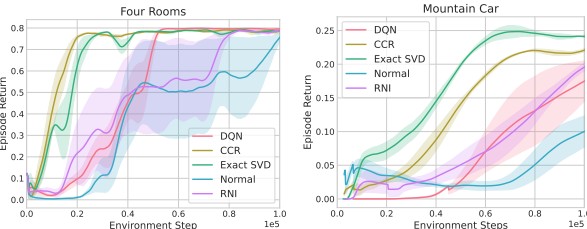

Figure 6: Comparing effects of offline pre-training on the FourRooms (**left**) and sparse Mountain Car (**right**) domains for different cumulant generation methods. Results are averages over three seeds.

the various cumulant generation methods and the effects of their corresponding GVFs as a representation pre-training method for reinforcement learning. Details can be found in Appendix A.

We consider four cumulant functions. The first two are stationary and are generated before offline pre-training begins. For ExactSVD, we compute the top-k right singular vectors of the successor representation matrix of the uniform random policy. For Normal, we generate cumulant functions sampled from a standard Normal distribution.

The second two cumulant functions are learned during offline pre-training using a separate neural network. RNI (Farebrother et al., 2023), learns a set of indicator functions which are trained to be active in a particular percentage of the states (15% in this experiment). Clustering Contrastive Representations (CCR) learns cumulants by learning a representation of the state using CPC (Oord et al., 2018), and then performs online clustering of the learned representations with $k$ clusters. The online clustering method we use differs slightly from standard approaches in that we maintain an estimate of the frequency that each cluster center is assigned to a state, $p_i$, and the assigned cluster is identified with $\arg\min_i p_i \|\phi(x) - b_i\|$, where $\phi(x)$ is the learned CPC representation and $b_i$ is cumulant $i$'s centroid. Examples of the cumulants produced by these four methods, and their corresponding value functions, are given in Appendix A.

Figure 6 compares the online performance after pre-training, for various cumulant functions, with the online performance of DQN without pre-training. Two take-aways are readily apparent. First, that offline pre-training, speeds up online learning, as expected. Second, that the two best performing methods are both sensitive to the structure of the environment dynamics, directly in the case of ExactSVD and indirectly through the CPC representation for CCR.

## 6. Related Work

**Optimal representations.** Bellemare et al. (2019) define a notion of optimal representations for batch Monte Carlo optimization based on the worst approximation error of the value function across the set of all possible policies, later relaxed by Dabney et al. (2021). Instead, we do not consider the control setting but focus on policy evaluation. Ghosh and Bellemare (2020) and Le Lan et al. (2022b) characterize the stability, approximation and generalization errors of the SR (Dayan, 1993) and Schur representations which are $P^\pi$-invariant, a key property to ensure stability. In contrast, we formalize that predicting values functions by TD learning from $G = I$ leads to $P^\pi$-invariant subspaces.

**Auxiliary tasks.** Lyle et al. (2021) analyse the representations learnt by several auxiliary tasks such as random cumulants (Osband et al., 2018; Dabney et al., 2021) assuming real diagonalizability of the transition matrix $P^\pi$ and constant weights $W$. They found that in the limit of an infinity of gaussian random cumulants, the subspace spanned by TD representations converges in distribution towards the left singular vectors of the successor representation. Instead, our theoretical analysis holds for any transition matrix and both the weights $W$ and the features $\Phi$ are updated at each time step. Recently, Farebrother et al. (2023) rely on a random binary cumulant matrix which sparsity is controlled by means of a quantile regression loss. Finally, other auxiliary tasks regroup self-supervised learning methods (Schwarzer et al., 2020; Guo et al., 2020). Tang et al. (2022) demonstrate that these algorithms perform an eigendecompositon of real diagonalisable transition matrix $P^\pi$, under some assumptions, suggesting a close connection to TD auxiliary tasks.

## 7. Conclusion

In this paper, we have studied representations learnt by bootstrapping methods and proved their benefit for value-based deep RL agents. Based on an analysis of the TD continuous-time dynamical system, we generalized existing work (Lyle et al., 2021) and provided evidence that TD representations are actually different from Monte Carlo representations.

Our investigation demonstrated that an identity cumulant matrix provides as much information as the TD and supervised auxiliary algorithms can carry; this work also shows that it is possible to design more compact pseudo-reward functions, though this requires prior knowledge about the transition dynamics. This led us to propose new families of cumulants which also proved useful empirically.

We assumed in this paper that the TD updates are carried out in tabular way, that is that there is not generalization between states when we update the features. An exciting opportunity for future work is to extend the theoretical results to the case where the representation is parametrized by a neural network. Other avenues for future work include scaling up the representation learning methods here introduced.

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

# A. Additional Empirical Results

### A.1. Additional details for Subsection 5.1

In this experiment, we selected a step size $\alpha = 0.08$ for all the algorithms. We also choose a uniform data distribution $\Xi = I/|\mathcal{S}|$ and a cumulant matrix $G = I$ for simplicity.

### A.2. Additional details for Subsection 5.2

In this experiment, we use a step size $\alpha = 5e - 3$ and train the different learning rules for 500k steps with 3 seeds. We consider the transition matrices induced by an epsilon greedy policy on the four room domain (Sutton et al., 1999) with $\epsilon = 0.8$ and train the supervised and TD update rules as described in Subsection 5.1

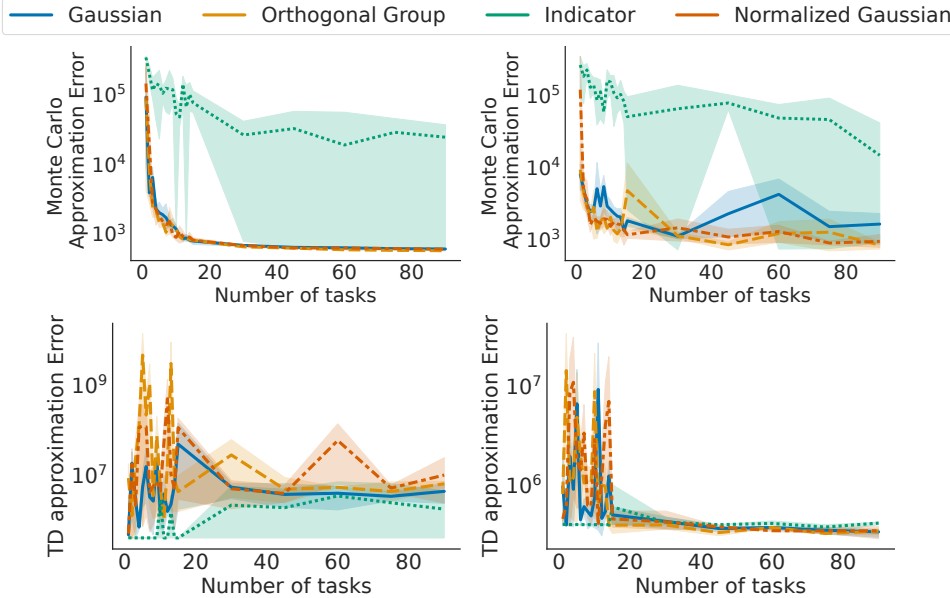

Figure 7: Monte Carlo and TD approximation errors after $5.10^5$ training steps on the learning rules $\mathcal{L}_{\text{aux}}^{\text{MC}}$ (on the **left** column) and $\mathcal{L}_{\text{aux}}^{\text{TD}}$ (on the **right** column) in the four-room domain for different distributions of cumulant, averaged over 3 seeds, for $d = 5$. Shaded areas represent $95\%$ confidence intervals.

### A.3. Additional details for Subsection 5.3

Four Rooms is a tabular gridworld environment where the agent begins in a room in the top left corner and must navigate to the goal state in the lower right corner. The actions are up, down, left and right and have deterministic effects. The reward function is one upon transitioning into the goal state and zero otherwise.

Mountain Car is a two-dimensional continuous state environment where the agent must move an under-powered car from the bottom of a valley to a goal state at the top of the nearby hill. The agent observes the continuous-valued position and velocity of the car, and controls it with three discrete actions which apply positive, negative, and zero thrust to the car. In this sparse reward version of the domain the reward is one for reaching the goal and zero otherwise. In this domain, we compute the ExactSVD by first discretizing the state space into approximately 2000 states, and compute an approximate $P^{\pi}$ by simulating transitions from uniformly random continuous states belonging to each discretized state.

In this evaluation we first pre-train a network representation offline with a large fixed dataset produced from following the uniform random policy. During offline pre-training the agent does not observe the reward, and instead learns action-value functions, GVFs, for each of several cumulant functions. After pre-training, the GVF head is removed and replaced with a single action-value function head. This network is then trained online with DQN on the true environmental reward. Note that we allow gradients to propagate into the network representation during online training.

In Four Rooms all methods use $k = 40$ cumulants and in Mountain Car all methods use $k = 80$ cumulants.

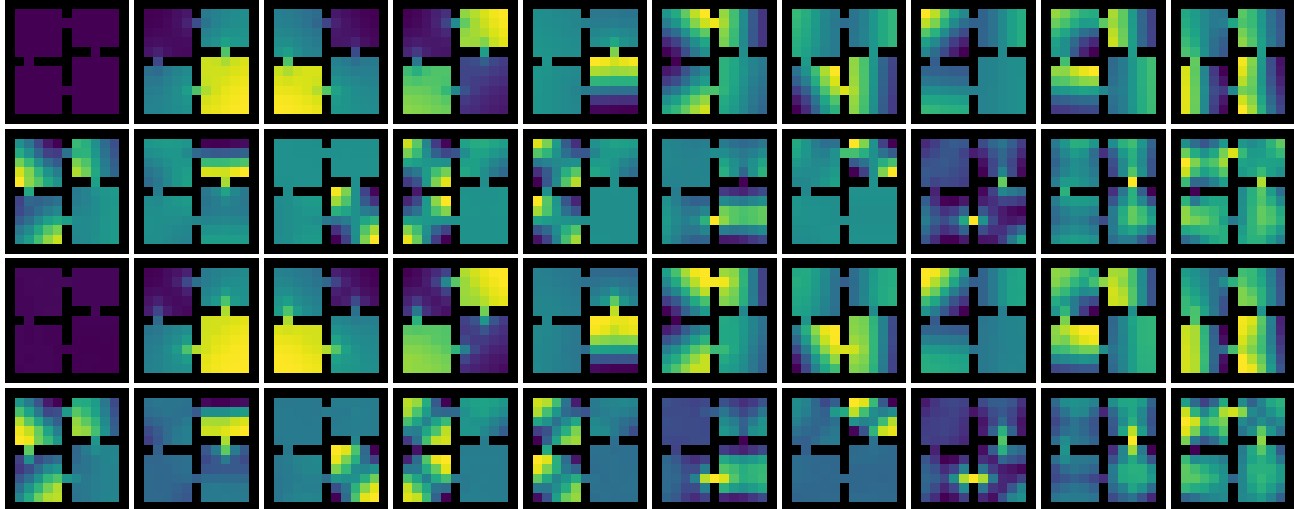

Figure 8: Example for ExactSVD of the learned cumulants (first two rows) and value functions (last two rows) during offline pre-training in FourRooms under the uniform random policy.

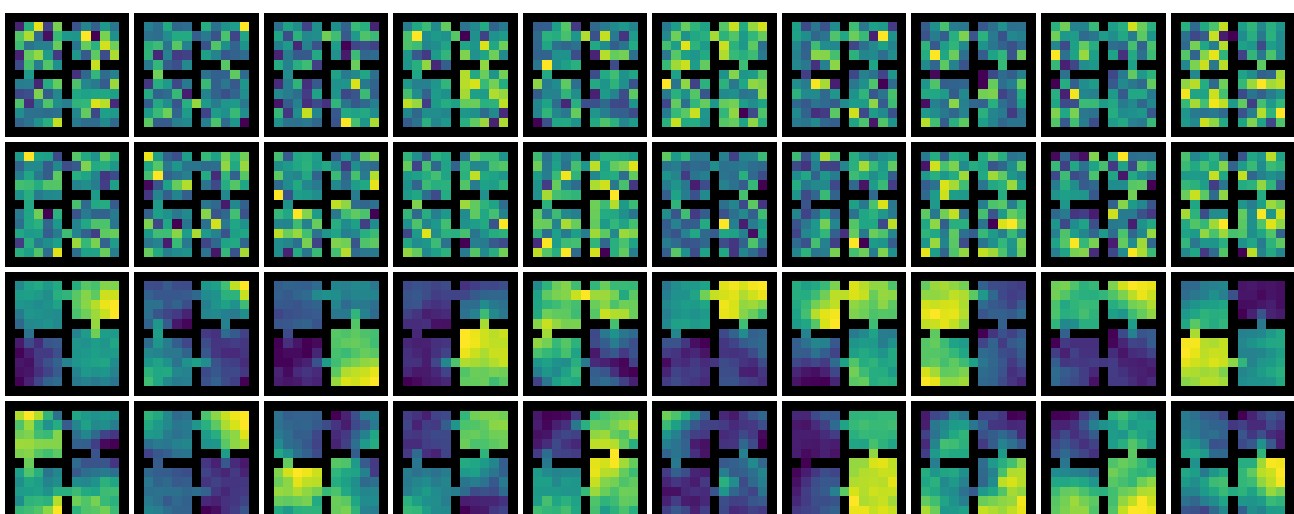

Figure 9: Example for Normal of the learned cumulants (first two rows) and value functions (last two rows) during offline pre-training in FourRooms under the uniform random policy.

## B. Proofs for Section 3

Throughout the appendix, we will use the notation $L := I - \gamma P^\pi$.

The beginning of this section is dedicated to proving the main result of Section 3, Theorem 1. Before that, we introduce the following necessary lemma.

**Lemma 7.** *Let $\Phi \in \mathbb{R}^{S \times d}$ and $\Psi \in \mathbb{R}^{S \times T}$. Let $P_\Phi$ be a (possibly oblique) projection onto $\mathrm{span}(\Phi)$. We have*

$$P_\Phi \Psi = \Psi \iff \mathrm{span}(\Psi) \subseteq \mathrm{span}(\Phi)$$

*Proof.* $P_\Phi$ can be written as $P_\Phi = \Phi(X^\top \Phi)^{-1} X^\top$ where $\Phi, X \in \mathbb{R}^{S \times d}$ and $X^\top \Phi \in \mathbb{R}^{d \times d}$ is invertible. Write $P_\Phi = \Phi Q$ with $Q = (X^\top \Phi)^{-1} X^\top$.

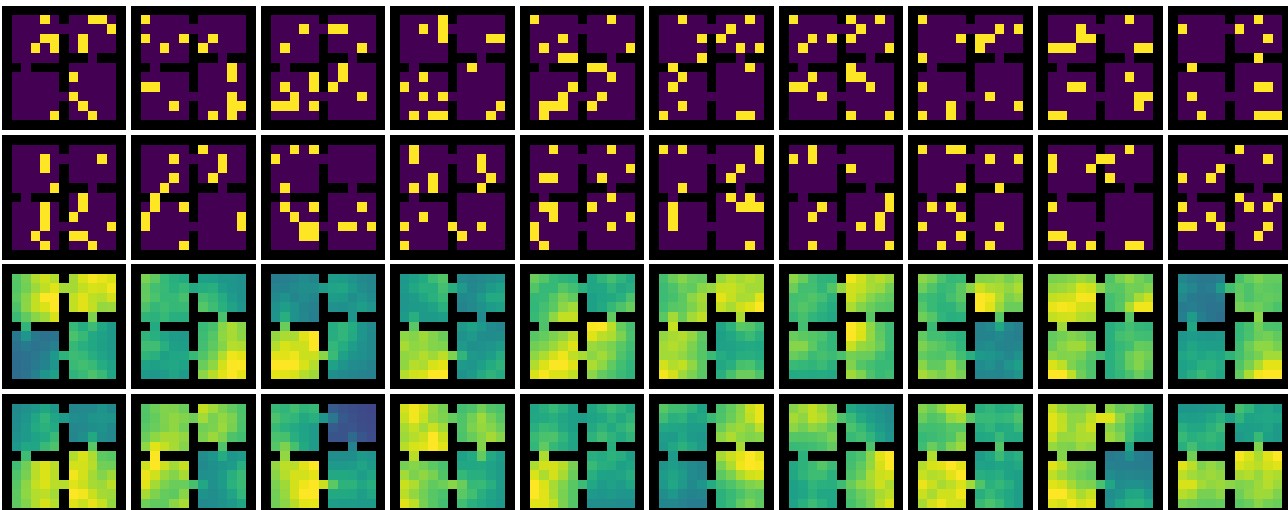

Figure 10: Example for CCR of the learned cumulants (first two rows) and value functions (last two rows) during offline pre-training in FourRooms under the uniform random policy.

Figure 11: Example for RNI of the learned cumulants (first two rows) and value functions (last two rows) during offline pre-training in FourRooms under the uniform random policy.

( $\implies$ ) Suppose $\Psi \in \mathbb{R}^{S \times T}$ such that $P_\Phi \Psi = \Psi$. Then, $\Psi = \Phi(Q\Psi)$. Let $\omega \in \mathbb{R}^T$. $\Psi\omega = \Phi(Q\Psi)\omega$ so $\Psi\omega \in \mathrm{span}(\Phi)$ Hence $\mathrm{span}(\Psi) \subseteq \mathrm{span}(\Phi)$.

( $\impliedby$ ) Suppose $\mathrm{span}(\Psi) \subseteq \mathrm{span}(\Phi)$. Denote $(e_t)$ the standard basis. We have $P_\Phi \Psi = (\sum_t P_\Phi(\Psi e_t)e_t^\top)$. Note that $\Psi e_t \in \mathrm{span}(\Psi) \subseteq \mathrm{span}(\Phi)$. Hence, there exists $y_t \in \mathbb{R}^d$ such that $\Psi e_t = \Phi y_t$. Now, $P_\Phi \Psi = (\sum_t P_\Phi(\Phi y_t)e_t^\top) = (\sum_t \Phi(X^\top \Phi)^{-1} X^\top \Phi y_t e_t^\top) = (\sum_t \Phi y_t e_t^\top) = (\sum_t \Psi e_t e_t^\top) = \Psi$. □

**Lemma 1** (Critical representations for TD). *All full rank representations which are critical points to $\mathcal{L}_{\mathrm{aux}}^{\mathrm{TD}}$ span real invariant subspaces of $(I - \gamma P^\pi)^{-1} GG^\top \Xi$, that is $\mathrm{span}((I - \gamma P^\pi)^{-1} GG^\top \Xi \Phi) \subseteq \mathrm{span}(\Phi)$.*

*Proof.* Start with these equations.

$$\text{For a fixed } \Phi, \nabla_W \|(\Xi)^{\frac{1}{2}} (\Phi W - G - \gamma P^\pi \mathrm{SG}[\Phi W])\|_F^2 = 2\Phi^\top \Xi(\Phi W - G - \gamma P^\pi \Phi W)$$

$$\text{For a fixed } W, \nabla_\Phi \|(\Xi)^{\frac{1}{2}} (\Phi W - G - \gamma P^\pi \mathrm{SG}[\Phi W])\|_F^2 = 2\Xi(\Phi W - G - \gamma P^\pi \Phi W)W^\top$$

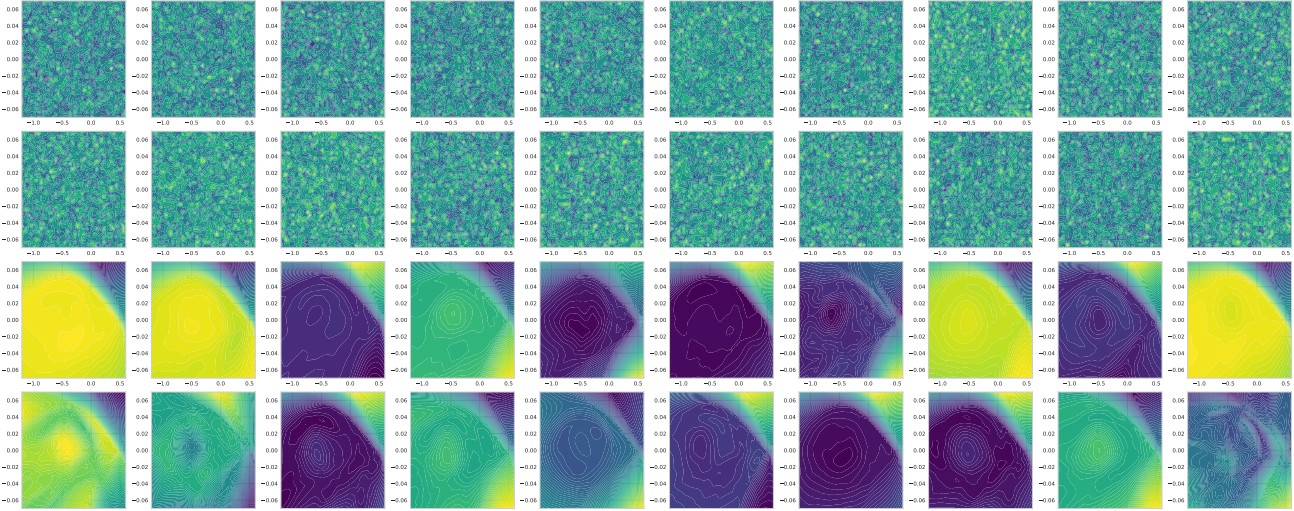

Figure 12: Example for ExactSVD of the learned cumulants (first two rows) and value functions (last two rows) during offline pre-training in sparse Mountain Car under the uniform random policy.

Figure 13: Example for Normal of the learned cumulants (first two rows) and value functions (last two rows) during offline pre-training in sparse Mountain Car under the uniform random policy.

By Assumption 1, $\Phi^\mathsf{T}\Xi L\Phi$ is invertible for all full rank representations $\Phi$. Hence, for a fixed full rank $\Phi$,

$$\nabla_W \|(\Xi)^{\frac{1}{2}}(\Phi W - G - \gamma P^\pi \mathrm{SG}[\Phi W])\|_F^2 = 0 \iff W_\Phi^* = \left(\Phi^\mathsf{T}\Xi L\Phi\right)^{-1}\Phi^\mathsf{T}\Xi G$$

Using the second fixed-point equation:

$$0 = (L\Phi W - G)W^\mathsf{T} \iff L\Phi WW^\mathsf{T} = GW^\mathsf{T}.$$

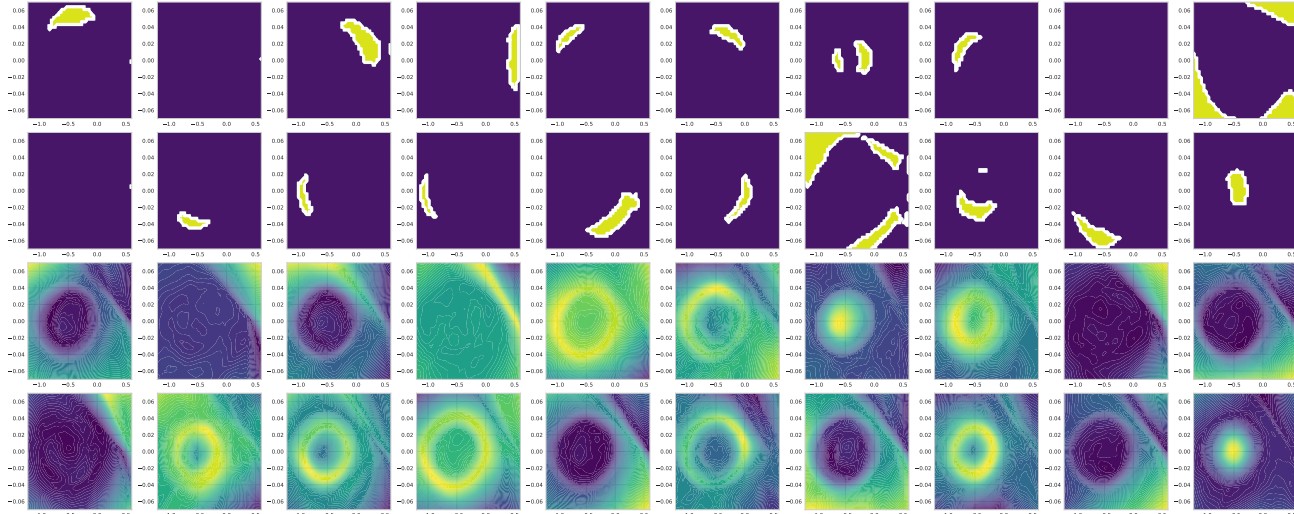

Figure 14: Example for CCR of the learned cumulants (first two rows) and value functions (last two rows) during offline pre-training in sparse Mountain Car under the uniform random policy.

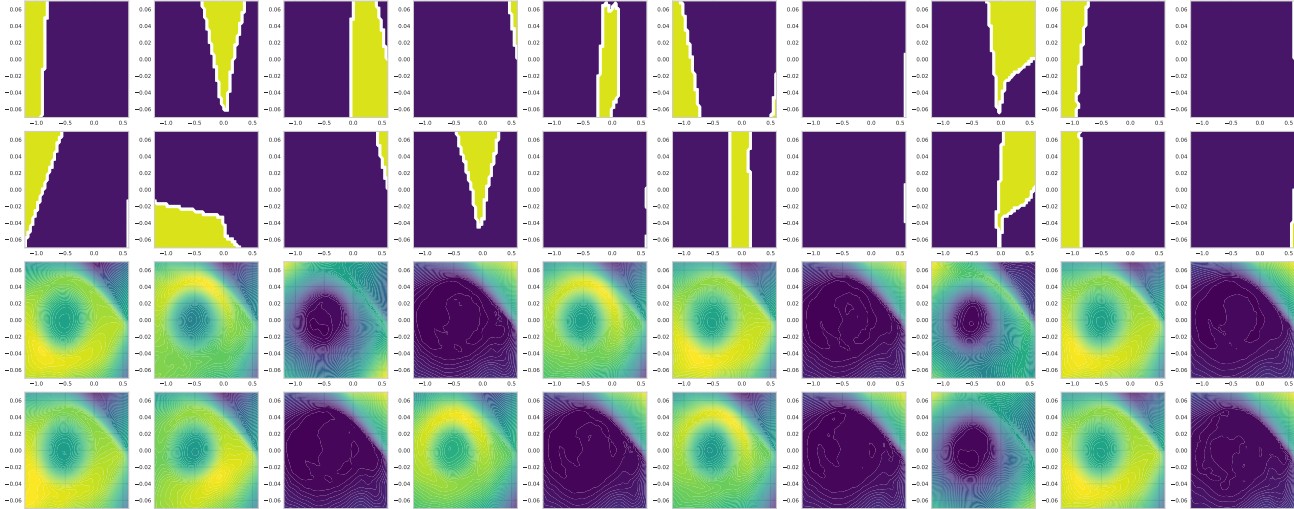

Figure 15: Example for RNI of the learned cumulants (first two rows) and value functions (last two rows) during offline pre-training in sparse Mountain Car under the uniform random policy.

Now plugging in the expression for $W_\Phi^*$,

$$L\Phi\left(\Phi^\mathsf{T} D_\pi L\Phi\right)^{-1}\Phi^\mathsf{T} D_\pi G\left(\left(\Phi^\mathsf{T} D_\pi L\Phi\right)^{-1}\Phi^\mathsf{T} D_\pi G\right)^\mathsf{T} = G\left(\left(\Phi^\mathsf{T} D_\pi L\Phi\right)^{-1}\Phi^\mathsf{T} D_\pi G\right)^\mathsf{T}$$

$$\Leftrightarrow L\Phi\left(\Phi^\mathsf{T} D_\pi L\Phi\right)^{-1}\Phi^\mathsf{T} D_\pi GG^\mathsf{T} D_\pi \Phi\left(\Phi^\mathsf{T} D_\pi L\Phi\right)^{-\mathsf{T}} = GG^\mathsf{T} D_\pi \Phi\left(\Phi^\mathsf{T} D_\pi L\Phi\right)^{-\mathsf{T}}$$

$$\Leftrightarrow \Phi\left(\Phi^\mathsf{T} D_\pi L\Phi\right)^{-1}\Phi^\mathsf{T} D_\pi GG^\mathsf{T} D_\pi \Phi = L^{-1}GG^\mathsf{T} D_\pi \Phi$$

$$\Leftrightarrow \Pi_{L^\mathsf{T} D_\pi \Phi} L^{-1}GG^\mathsf{T} D_\pi \Phi = L^{-1}GG^\mathsf{T} D_\pi \Phi$$

where $\Pi_X = \Phi(X^\mathsf{T}\Phi)^{-1}X^\mathsf{T}$ is the oblique projection onto $\mathrm{span}(\Phi)$ orthogonally to $\mathrm{span}(X)$. This is equivalent to $\Pi_{L^\mathsf{T} D_\pi \Phi}^\perp L^{-1}GG^\mathsf{T} D_\pi \Phi = 0$, which is equivalent to saying that $\mathrm{span}(\Phi)$ must be an *invariant subspace* of $L^{-1}GG^\mathsf{T} D_\pi$ by Lemma 7.

In other words, we have shown that all non-degenerate full-rank $\Phi$ which are critical points span invariant subspaces of $L^{-1}GG^\mathsf{T}D_\pi$. We can enumerate these via the *real Jordan normal form* of $L^{-1}GG^\mathsf{T}D_\pi$. Each block of the real Jordan normal form corresponds to an invariant subspace of $L^{-1}GG^\mathsf{T}D_\pi$. Suppose that the real Jordan block sizes of $L^{-1}GG^\mathsf{T}D_\pi$ are $n_1, n_2, ..., n_b$ ($L^{-1}GG^\mathsf{T}D_\pi$ has $b$ real Jordan blocks), and suppose $L^{-1}GG^\mathsf{T}D_\pi = SJS^{-1}$ is the real Jordan decomposition, with $J = \text{blkdiag}(J_{n_1}(\lambda_1), ..., J_{n_b}(\lambda_b))$. Partition the columns of $S$ into $S_1, ..., S_b$. Then if $\Phi \in \mathbb{R}^{S\times k}$, the set of non-degenerate stationary full rank representations is:

$$\{\begin{bmatrix} S_{i_1} & ... & S_{i_\ell} \end{bmatrix} \mid n_{i_1} + ... + n_{i_\ell} = k\}.$$

$\square$

**Corollary 1.** *If $G = I$ and $\Xi = I/|S|$, all full rank representations which are critical points to $\mathcal{L}^{\text{TD}}_{\text{aux}}$ span real invariant subspaces of the invariant subspaces of $P^\pi$.*

*Proof.* By Theorem 1, all full rank representations which are critical points of $\mathcal{L}^{\text{TD}}_{\text{aux}}$ span real invariant subspaces of $(I - \gamma P^\pi)^{-1}$.

Let $\Phi$ be a representation spanning an invariant subspace of $(I - \gamma P^\pi)^{-1}$. By definition, $\text{span}((I-\gamma P^\pi)^{-1}\Phi) \subseteq \text{span}(\Phi)$. Because $(I-\gamma P^\pi)$ is invertible, we have $\dim((I-\gamma P^\pi)^{-1}\Phi) = \dim(\Phi)$. Hence, we actually have $\text{span}((I-\gamma P^\pi)^{-1}\Phi) = \text{span}(\Phi)$. There exists $w_1, w_2 \in \mathbb{R}^d$ such that $\Phi w_1 = (I - \gamma P^\pi)^{-1}\Phi w_2$ so $(I - \gamma P^\pi)\Phi w_1 = \Phi w_2$. It follows that $\Phi \frac{(w_1 - w_2)}{\gamma} = P^\pi \Phi w_1$. Hence, $P^\pi \Phi w_1 \in \text{span}(\Phi)$ and $\text{span}(P^\pi\Phi) \subseteq \text{span}(\Phi)$. We conclude that $\Phi$ spans an invariant subspace of $P^\pi$. $\square$

**Theorem 1** (TD representations). *Assume $G = I$ and a uniform distribution $\xi$ over states. Let $\lambda_1, .., \lambda_{|S|}$ be the (possibly complex) eigenvalues of $P^\pi$, ordered by decreasing real part $\text{Re}(\lambda_i) \geqslant \text{Re}(\lambda_{i+1})$, $i \in \{1, .., |\mathcal{S}|\}$. Under the dynamics in Equation (2), all real invariant subspaces of dimension $d$ are critical points, and only the top-$d$ invariant subspace, if it exists, is stable.*

*Proof.* Consider this objective:

$$\mathcal{L}(\Phi) = \frac{1}{2}\|(\Xi^{\frac{1}{2}})(\Phi W^{\text{TD}}_{\Phi,G} - G - \gamma P^\pi \text{SG}[\Phi W^{\text{TD}}_{\Phi,G}])\|^2_F,$$

and $W^{\text{TD}}_{\Phi,G} = (\Phi^\mathsf{T}\Xi L\Phi)^{-1}\Phi^\mathsf{T}\Xi G$ and define $L := I - \gamma P^\pi$. Observe that:

$$\text{For a fixed } W, \nabla_\Phi \|\Phi W - G - \gamma P^\pi \text{SG}[\Phi W]\|^2_F = 2\Xi(L\Phi W^*_\Phi - G)(W^*_\Phi)^\mathsf{T}$$

So now we consider the continuous time dynamics:

$$\frac{d}{dt}\Phi = -\nabla_\Phi \mathcal{L}(\Phi) := -F(\Phi), \tag{5}$$

where:

$$F(\Phi) := \Xi(L\Phi W^{\text{TD}}_{\Phi,G} - G)(W^{\text{TD}}_{\Phi,G})^\mathsf{T} = \Xi L(\Pi_{L^\mathsf{T}\Xi\Phi} - I)L^{-1}GG^\mathsf{T}\Xi\Phi(\Phi^\mathsf{T}\Xi L\Phi)^{-\mathsf{T}}$$

Consider the case $G = I$ and $\Xi = I/|\mathcal{S}|$. The proof strategy consists in constructing an eigenvector $\Delta \in \mathbb{R}^{S\times d}$ of $\partial_\Phi F(\Phi)$ as a function of $\Phi, L, G$ such that $\partial_\Phi F(\Phi)[\Delta] = -\lambda\Delta$ for some $\text{Re}(\lambda) > 0$. For every non top-$d$ invariant subspace, we prove that the Jacobian of the dynamics $-F$ has a positive real part eigenvalue.

Let $\Phi$ be a stationary point which columns are orthogonal such that $\Phi^\mathsf{T}\Phi = Id$. $\Phi$ is an invariant subspace of $P^\pi$. Assume that $\Phi$ does not contain the eigenvectors corresponding to the top $d$ eigenvalues. Define $\Lambda = \text{diag}(\lambda_1, ..., \lambda_d)$ its associated eigenvalues assumed distinct. We have $P\Phi = \Phi\Lambda$. Hence, $(I - \gamma P^\pi)\Phi = \Phi(I - \gamma\Lambda)$. Let $\lambda_{\max}$ the largest eigenvalue of $P^\pi$ not contained in $\Phi$ and let $i \in \{1, .., d\}$ be the largest index such that $\lambda_i < \lambda_{\max}$. Let $\Delta$ be the matrix with the largest eigenvector $\lambda_{\max}$ of $P^\pi$ not contained in $\Phi$ in its $i$-th column and 0 elsewhere.

$$\partial_\Phi W^*_\Phi[\Delta] = -(\Phi^\mathsf{T}L\Phi)^{-1}(\Delta^\mathsf{T}L\Phi + \Phi^\mathsf{T}L\Delta)(\Phi^\mathsf{T}L\Phi)^{-1}\Phi^\mathsf{T}G + (\Phi^\mathsf{T}L\Phi)^{-1}\Delta^\mathsf{T}G$$
$$= -(\Phi^\mathsf{T}\Phi(I_d - \gamma\Lambda))^{-1}(\Delta^\mathsf{T}\Phi(I - \gamma\Lambda) + (1 - \gamma\lambda_{\max})\Phi^\mathsf{T}\Delta)(\Phi^\mathsf{T}\Phi(I_d - \gamma\Lambda))^{-1}\Phi^\mathsf{T}G + (\Phi^\mathsf{T}\Phi(I_d - \gamma\Lambda))^{-1}\Delta^\mathsf{T}$$
$$= (I_d - \gamma\Lambda)^{-1}(\Phi^\mathsf{T}\Phi)^{-1}\Delta^\mathsf{T}$$

$$\partial_\Phi F(\Phi)[\Delta] = (L\Delta W_\Phi^* + L\Phi(dW_\Phi^*))(W_\Phi^*)^\mathsf{T} + (L\Phi W_\Phi^* - G)(dW_\Phi^*)^\mathsf{T}$$

$$= (1 - \gamma\lambda_{\max})\Delta(I - \gamma\Lambda)^{-2}(\Phi^\mathsf{T}\Phi)^{-1} + L\Phi(I - \gamma\Lambda)^{-1}(\Phi^\mathsf{T}\Phi)^{-1}\Delta^\mathsf{T}\Phi(\Phi^\mathsf{T}L\Phi)^{-\mathsf{T}}$$

$$+ L\Phi(\Phi^\mathsf{T}L\Phi)^{-1}\Phi^\mathsf{T}\Delta(\Phi^\mathsf{T}\Phi)^{-1}(I - \gamma\Lambda)^{-\mathsf{T}} - \Delta(\Phi^\mathsf{T}\Phi)^{-1}(I - \gamma\Lambda)^{-\mathsf{T}}$$

$$= \Delta(1 - \gamma\lambda_{\max})(I - \gamma\Lambda)^{-2}(\Phi^\mathsf{T}\Phi)^{-1} - \Delta(\Phi^\mathsf{T}\Phi)^{-1}(I - \gamma\Lambda)^{-\mathsf{T}}$$

$$= \Delta(1 - \gamma\lambda_{\max})(I - \gamma\Lambda)^{-2} - \Delta(I - \gamma\Lambda)^{-1}$$

$$= \gamma\Delta(-\lambda_{\max}I + \Lambda)(I - \gamma\Lambda)^{-2}$$

$$= \gamma\Delta(-\lambda_{\max} + \lambda_i)(1 - \gamma\lambda_i)^{-2} < 0$$

Hence, any non top-$d$ invariant subspace is unstable for gradient descent. □

**Proposition 2** (Residual representations). *Let $d \in \{1, ..., S\}$ and $F_d$ be the top $d$ left singular vectors of $G$ with respect to the inner product $\langle x, y\rangle_\Xi = y^\mathsf{T}\Xi x$, for all $x, y \in \mathbb{R}^{|S|}$. All representations spanning $(I - \gamma P^\pi)^{-1}F_d$ are global minimizers of $\mathcal{L}_{\text{aux}}^{\text{res}}$ and can be recovered by stochastic gradient descent.*

*Proof.* We can write the loss function to be minimized as

$$J(\Phi) = \min_{W \in \mathbb{R}^{d \times T}} \|\Xi^{1/2}(\Phi W - (G + \gamma P^\pi \Phi W))\|_F^2$$

$$= \min_{W \in \mathbb{R}^{d \times T}} \|\Xi^{1/2}(\Phi W - \gamma P^\pi \Phi W - G)\|_F^2$$

$$= \min_{W \in \mathbb{R}^{d \times T}} \|\Xi^{1/2}((I - \gamma P^\pi)\Phi W - G)\|_F^2$$

Now,

$$\arg\min_{\Phi \in \mathbb{R}^{S \times d}} \min_{W \in \mathbb{R}^{d \times T}} \|\Xi^{1/2}((I - \gamma P^\pi)\Phi W - G)\|_F^2 = \arg\min_{\Phi \in \mathbb{R}^{S \times d}} \|P_{\Xi^{1/2}(I-\gamma P^\pi)\Phi}^\perp \Xi^{1/2}G\|_F^2$$

$$= \{\Phi \in \mathbb{R}^{S \times d} \mid \Phi = (I - \gamma P^\pi)^{-1}F_d M, M \in GL_d(\mathbb{R})\}$$

This set of representations can be recovered by stochastic gradient descent efficiently, i.e., with number of SGD iterations scaling at most polynomially in all problem specific parameters (Ge et al., 2017; Jin et al., 2017) in the context of SGD. □

**Proposition 1** (Monte Carlo representations). *If $\text{rank}(\Psi^\pi) \geqslant d$, all representations spanning the top-$d$ left singular vectors of $\Psi^\pi$ with respect to the inner product $\langle x, y\rangle_\Xi$ are global minimizers of $\mathcal{L}_{\text{aux}}^{\text{MC}}$ and can be recovered by stochastic gradient descent.*

*Proof.* Let $F_d$ denote the top $d$ left singular vectors of $\Psi$.

$$\arg\min_{\Phi \in \mathbb{R}^{S \times d}} \min_{W \in \mathbb{R}^{d \times T}} \|\Xi^{1/2}(\Phi W - \Psi)\|_F^2 = \arg\min_{\Phi \in \mathbb{R}^{S \times d}} \|P_{\Xi^{1/2}\Phi}^\perp \Xi^{1/2}\Psi\|_F^2$$

$$= \{\Phi \in \mathbb{R}^{S \times d} \mid \exists M \in GL_d(\mathbb{R}), \Phi = F_d M\}$$

This set of representations can be recovered by stochastic gradient descent efficiently, i.e., with number of SGD iterations scaling at most polynomially in all problem specific parameters (Ge et al., 2017; Jin et al., 2017) in the context of SGD. □

**Corollary 2** (Symmetric transition matrices). *If a cumulant matrix $G \in \mathbb{R}^{S \times T}$ (with $T \geqslant S$) has unit-norm, orthogonal columns (e.g. $G = I$), the representations learnt from the supervised objective $\mathcal{L}_{\text{aux}}^{\text{MC}}$ and the TD update rule $\mathcal{L}_{\text{aux}}^{\text{TD}}$ are the same for symmetric transition matrices $P^\pi$ under a uniform state distribution $\xi$.*

*Proof.* Assume that $P^\pi$ is symmetric so that $L$ and $L^{-1}$ are also symmetric.

By Proposition 1, running SGD on the supervised objective $\mathcal{L}_{\text{aux}}^{\text{MC}}$ using $\Psi = L^{-1}G$ as targets results in a representation spanning the top-$d$ left singular vectors of $L^{-1}G$ which are the same as the top-$d$ left singular vectors of $L^{-1}$.

By assumption $G$ is orthogonal, hence $GG^\mathsf{T} = I$. Because $L^{-1}GG^\mathsf{T}$ is symmetric, all its eigenvalues are real. By Theorem 1, running gradient descent on $\mathcal{L}_{\text{aux}}^{\text{TD}}$ using $G$ as the cumulant matrix converges to the top-$d$ eigenvectors of

990 $L^{-1}GG^{\mathsf{T}} = L^{-1}$. Indeed, the subspaces given by the span of the right eigenvectors of $L^{-1}$ are the only $L^{-1}$-invariant
991 subspaces. These eigenvectors are also the singular vectors of $L^{-1}$ as this matrix is symmetric.

992
993 Because $P$ is a row stochastic matrix, we have that the spectral radius of $P$ satisfies $\rho(P) = 1$, and therefore $\lambda(P) \subseteq [-1, 1]$.
994 Hence:

995
996 $$\frac{1}{1 - \gamma\lambda} \in [1/(1+\gamma), 1/(1-\gamma)].$$

997
998 Hence, the eigenvalues of $L^{-1}$ are positive. Because $L^{-1}$ is symmetric, the singular values of $L^{-1}$ are exactly its eigenvalues.
999 Hence, the top-$d$ eigenvectors are the top-$d$ singular vectors and the conclusion follows. □

1000

## C. Proofs for Section 4

1001
1002
1003 **Lemma 2.** *A representation $\Phi^*_{\mathrm{TD}}$ is $l_1$-ball optimal for TD learning iff it is a solution of the following optimization problem.*

1004
1005 $$\Phi^*_{\mathrm{TD}} \in \arg\min_{\Phi} \left\| \Xi^{1/2}(\Phi W^{\mathrm{TD}}_{\Phi,I} - (I - \gamma P^\pi)^{-1}) \right\|^2_F.$$

1006
1007 *Proof.* By definition, a representation is enough for TD learning when it is a minimizer of Equation (3), that is,

1008
1009 $$\Phi^*_{\mathrm{TD}} \in \arg\min_{\Phi \in \mathbb{R}^{S \times d}} \mathbb{E}_{r_\pi} \|\Phi w^{\mathrm{TD}}_\Phi - V^\pi\|^2_\xi, \tag{6}$$

1010
1011 where the expectation is over the reward functions $r_\pi$ sampled uniformly over the $l_1$ ball $\|r_\pi\|^2_1 \leqslant 1$ and
1012

1013 $$w^{\mathrm{TD}}_\Phi = \left(\Phi^{\mathsf{T}}\Xi(I - \gamma P^\pi)\Phi\right)^{-1}\Phi^{\mathsf{T}}\Xi r_\pi.$$
1014

1015 Write $P^\perp_{L^{\mathsf{T}}\Xi\Phi} = I - P_{L^{\mathsf{T}}\Xi\Phi}$ and $P_X = \Phi(X^{\mathsf{T}}\Phi)^{-1}X^{\mathsf{T}}$ the oblique projection onto $\mathrm{span}(\Phi)$ orthogonally to $\mathrm{span}(X)$. We
1016 have
1017

1018 $$\mathbb{E}_{\|r\|^2_1 \leqslant 1}\|\Phi w^{\mathrm{TD}}_\Phi - V^\pi\|^2_\xi = \mathbb{E}_{\|r\|^2_1 \leqslant 1}\|\Xi^{1/2}P^\perp_{L^{\mathsf{T}}\Xi\Phi}(I - \gamma P^\pi)^{-1}r\|^2_2$$
1019
1020 $$= \mathbb{E}_{\|r\|^2_1 \leqslant 1}\|\Xi^{1/2}P^\perp_{L^{\mathsf{T}}\Xi\Phi}(I - \gamma P^\pi)^{-1}r\|^2_2$$
1021
1022 $$= \mathbb{E}_{\|r\|^2_1 \leqslant 1}\mathrm{tr}(r^{\mathsf{T}}L^{-\mathsf{T}}P^\perp_{L^{\mathsf{T}}\Xi\Phi}\Xi P^\perp_{L^{\mathsf{T}}\Xi\Phi}L^{-1}r)$$
1023
1024 $$= \mathrm{tr}(L^{-\mathsf{T}}P^\perp_{L^{\mathsf{T}}\Xi\Phi}\Xi P^\perp_{L^{\mathsf{T}}\Xi\Phi}L^{-1}\mathbb{E}(rr^{\mathsf{T}}))$$
1025
1026 $$= \|\Xi^{1/2}P^\perp_{L^{\mathsf{T}}\Xi\Phi}L^{-1}\|^2_F$$
1027
1028 $$= \left\| \Xi^{1/2}(\Phi W^{\mathrm{TD}}_{\Phi,I} - (I - \gamma P^\pi)^{-1}) \right\|^2_F$$

□

1029
1030 **Lemma 3.** *A representation $\Phi^*_{\mathrm{MC}}$ is $l_1$-ball optimal for batch Monte Carlo policy evaluation if its column space spans the
1031 top-d left singular vectors (with respect to the inner product $\langle x, y\rangle_\Xi$) of $(I - \gamma P^\pi)^{-1}$.*

1032
1033 *Proof.* We have

1034
1035 $$\mathbb{E}_{\|r\|^2_1 \leqslant 1}\|\hat{V}^{\mathrm{MC}} - V^\pi\|^2_\xi = \mathbb{E}_{\|r\|^2_1 \leqslant 1}\|P^\perp_{\Xi^{1/2}\Phi}\Xi^{1/2}(I - \gamma P^\pi)^{-1}r\|^2_2$$
1036
1037 $$= \mathbb{E}_{\|r\|^2_1 \leqslant 1}\mathrm{tr}(r^{\mathsf{T}}L^{-\mathsf{T}}\Xi^{1/2}P^\perp_{\Xi^{1/2}\Phi}\Xi^{1/2}L^{-1}r)$$
1038
1039 $$= \mathrm{tr}(L^{-\mathsf{T}}\Xi^{1/2}P^\perp_{\Xi^{1/2}\Phi}\Xi^{1/2}L^{-1}\mathbb{E}(rr^{\mathsf{T}}))$$
1040
1041 $$= \|P^\perp_{\Xi^{1/2}\Phi}\Xi^{1/2}L^{-1}\|^2_F$$

1042
1043 Write $(I - \gamma P^\pi)^{-1} = F\Sigma B^{\mathsf{T}}$ the weighted SVD of $(I - \gamma P^\pi)^{-1}$ where $F \in \mathbb{R}^{S \times S}$ such that $F^{\mathsf{T}}\Xi F = I$ and $B \in \mathbb{R}^{S \times S}$
1044 such that $B^{\mathsf{T}}B = I$. Write $F_d$ the top-$d$ left singular vectors corresponding to the top-$d$ singular values on the diagonal of $\Sigma$.

By definition, an $l_1$-ball optimal representation is solution to the following optimization problem

$$\underset{\Phi \in \mathbb{R}^{S \times d}}{\arg \min} \mathbb{E}_{\|r\|_1^2 \leqslant 1} \|\hat{V}^{\mathrm{MC}} - V^\pi\|_\xi^2 = \underset{\Phi \in \mathbb{R}^{S \times d}}{\arg \min} \|P_{\Xi^{1/2}\Phi}^\perp \Xi^{1/2} L^{-1}\|_F^2$$

$$= \underset{\Phi \in \mathbb{R}^{S \times d}}{\arg \min} \|P_{\Xi^{1/2}\Phi}^\perp \Xi^{1/2} F\Sigma B^\top\|_F^2$$

By the Eckart-Young theorem, $\|P_{F_d}^\perp \Xi^{1/2} F\Sigma B^T\|_F^2 \leqslant \|P_\Phi^\perp \Xi^{1/2} F\Sigma B^T\|_F^2$. Hence, the set of optimal representations is $\{F_d M, M \in GL_d(\mathbb{R})\}$. $\square$

**Lemma 8.** *Write $F_d \Sigma_d B_d^\top$ the truncated weighted SVD of the successor representation $(I - \gamma P^\pi)^{-1}$. A representation is $l_1$-ball optimal for residual policy evaluation if its column space spans $F_d \Sigma_d$.*

*Proof.* Write $(I - \gamma P^\pi)^{-1} = F\Sigma B^\top$ the weighted SVD of $(I - \gamma P^\pi)^{-1}$ where $F \in \mathbb{R}^{S \times S}$ such that $F^\top \Xi F = I$ and $B \in \mathbb{R}^{S \times S}$ such that $B^\top B = I$. Write $F_d$ the top-$d$ left singular vectors corresponding to the top-$d$ singular values on the diagonal of $\Sigma$. For a fixed $\Phi \in \mathbb{R}^{S \times d}$, the solution of $\min_{w \in \mathbb{R}^d} \|\Xi^{1/2}(\Phi w - (r_\pi + \gamma P^\pi \Phi w))\|_F^2$ is the Bellman residual minimizing approximation ([Lagoudakis and Parr, 2003]) and is given by

$$w_\Phi^{\mathrm{res}} = \left((\Phi - \gamma P^\pi \Phi)^\top \Xi (\Phi - \gamma P^\pi \Phi)\right)^{-1} (\Phi - \gamma P^\pi \Phi)^\top \Xi r_\pi.$$

Hence, the value approximant can be expressed by means of an orthogonal projection matrix as follows

$$\Phi w_\Phi^{\mathrm{res}} = (I - \gamma P^\pi)^{-1} \Xi^{-1/2} P_{\Xi^{1/2}(I - \gamma P^\pi)\Phi} \Xi^{1/2} r_\pi$$

where $P_X = X(X^\top X)^{-1} X^T$ denotes an orthogonal projection. By definition, a representation $l_1$-ball optimal for residual policy evaluation is solution to the following optimization problem

$$\underset{\Phi \in \mathbb{R}^{S \times d}}{\arg \min} \mathbb{E}_{\|r\|_1^2 \leqslant 1} \|\hat{V}^{\mathrm{res}} - V^\pi\|_\xi^2 = \underset{\Phi \in \mathbb{R}^{S \times d}}{\arg \min} \|\Xi^{1/2}(I - \gamma P^\pi)^{-1} \Xi^{-1/2} P_{\Xi^{1/2}(I - \gamma P^\pi)\Phi} \Xi^{1/2} r_\pi - \Xi^{1/2}(I - \gamma P^\pi)^{-1} r_\pi\|_F^2$$

$$= \underset{\Phi \in \mathbb{R}^{S \times d}}{\arg \min} \|\Xi^{1/2}(I - \gamma P^\pi)^{-1} \Xi^{-1/2} P_{\Xi^{1/2}(I - \gamma P^\pi)\Phi} \Xi^{1/2} - \Xi^{1/2}(I - \gamma P^\pi)^{-1}\|_F^2$$

$$= \underset{\Phi \in \mathbb{R}^{S \times d}}{\arg \min} \|\Xi^{1/2}(I - \gamma P^\pi)^{-1} P_{\Xi^{1/2}(I - \gamma P^\pi)\Phi}^\perp\|_F^2$$

Using an oblique projection,

$$\Phi w_\Phi^{\mathrm{res}} = (I - \gamma P^\pi)^{-1} \Xi^{-1/2} P_{\Xi^{1/2}(I - \gamma P^\pi)\Phi} \Xi^{1/2} r_\pi$$

$$\underset{\Phi \in \mathbb{R}^{S \times d}}{\arg \min} \mathbb{E}_{\|r\|_1^2 \leqslant 1} \|\hat{V}^{\mathrm{res}} - V^\pi\|_\xi^2 = \underset{\Phi \in \mathbb{R}^{S \times d}}{\arg \min} \|\Xi^{1/2}(I - \gamma P^\pi)^{-1} \Xi^{-1/2} P_{\Xi^{1/2}(I - \gamma P^\pi)\Phi} \Xi^{1/2} r_\pi - \Xi^{1/2}(I - \gamma P^\pi)^{-1} r_\pi\|_F^2$$

$$= \underset{\Phi \in \mathbb{R}^{S \times d}}{\arg \min} \|\Xi^{1/2}(I - \gamma P^\pi)^{-1} \Xi^{-1/2} P_{\Xi^{1/2}(I - \gamma P^\pi)\Phi} \Xi^{1/2} - \Xi^{1/2}(I - \gamma P^\pi)^{-1}\|_F^2$$

$$= \underset{\Phi \in \mathbb{R}^{S \times d}}{\arg \min} \|\Xi^{1/2}(I - \gamma P^\pi)^{-1} P_{\Xi^{1/2}(I - \gamma P^\pi)\Phi}^\perp\|_F^2$$

$$L^{-1} = U\Sigma V^\top$$

$L^{-1} \times$ the top $d$ right singular vectors of $(I - \gamma P^\pi)^{-1}$ is a solution. Let $U_d, \Sigma_d, V_d$ correspond to the top $d$ svals. Lets say that $U_d$ is $S \times d$, $\Sigma_d$ is square, and $V_d$ is also $S \times d$. What is $V^\top V_d = \begin{bmatrix} I_d \\ 0 \end{bmatrix}$.

---

What you want is $L\Phi = V_d$ so $\Phi = L^{-1} V_d = U\Sigma V^\top V_d = U_d \Sigma_d$. If $L\Phi = V_d$, then $P_{L\Phi}^\perp = P_{V_d}^\perp$, so $L^{-1} P_{V_d}^\perp = U_d^\perp \Sigma_d^\perp (V_d^\perp)^\top$, so the objective is now sum of the last $S - d$ singular values squared.

$\square$

# D. Proofs for Subsection 4.1

**Lemma 6.** *The set of cumulant matrices $G \in \mathbb{R}^{S \times T}$ that preserve the top-$T$ invariant subspaces of the successor representation by TD learning are the top-$T$ orthogonal invariant subspaces of $(I - \gamma P^\pi)^{-1}$, that is satisfying $G^\top G = I$ by orthogonality and $(I - \gamma P^\pi)^{-1} G \subseteq G$ by the invariance property.*

*Proof.* Let $\Phi \in \mathbb{R}^{S \times d}$ spanning an invariant subspace of $L^{-1}$. By definition, there exists a block diagonal matrix $J_\Phi \in \mathbb{R}^{d \times d}$ such that $L^{-1}\Phi = \Phi J_\Phi$. Let $G \in O(S, T)$ spanning the top $T$ invariant subspaces of $L^{-1}$. By definition, there exists a block diagonal matrix $J_G \in \mathbb{R}^{d \times d}$ such that $L^{-1}G = GJ_G$. Hencer, we have

$$(L^{-1}GG^\top)\Phi = (L^{-1}G)G^\top\Phi$$
$$= GJ_T G^\top \Phi$$
$$= (\Phi J_\Phi) \text{ by orthonormality}$$

Then, $\Phi$ is an invariant subspace of $L^{-1}GG^\top$. $\qquad\square$

**Lemma 5.** *Let $\{\Phi_\omega\}$ be the set of rotating representations from Figure 2 learnt by TD learning with $G = I$ and $d = 2$. All these representations are equally good for learning the main value function by TD learning, that is $\forall \omega \in [0, 1]$,*

$$\mathbb{E}_{\|r\|_2^2 < 1} \left\| \Phi_\omega w_{\Phi_\omega}^{\mathrm{TD}} - V^\pi \right\|_F^2$$

*is constant and independent of $\omega$.*

*Proof.* Let's start by considering the case of the three-state circular example. We consider an orthogonal basis for the invariant subspaces of $\Phi$. By definition, $P^\pi e_1 = e_1, P^\pi[e_2, e_3] = [e_2, e_3]\Lambda$ so $Le_1 = (1 - \gamma)e_1$ and $L[e_2, e_3] = (I - \gamma P)[e_2, e_3] = [e_2, e_3] - \gamma[e_2, e_3]\Lambda = [e_2, e_3](I - \gamma\Lambda)$.

Assume that there exists $\omega \in [0, 1]$ such that the representation is $\Phi = [e_1, \omega e_2 + (1 - \omega)e_3] = [e_1, e_2, e_3]\Omega$ with $\Omega = \begin{bmatrix} 1 & 0 \\ 0 & \omega \\ 0 & (1 - \omega) \end{bmatrix}$. $L\Phi = [(1 - \gamma)e_1, [e_2, e_3](I - \gamma\Lambda)]\Omega$. Hence, we have $L\Phi = [e_1, e_2, e_3]\begin{bmatrix} 1 - \gamma & 0 \\ 0 & I - \gamma\Lambda \end{bmatrix}\Omega$

and $\Phi^\top L\Phi = \Omega^\top[e_1, e_2, e_3]^\top[e_1, e_2, e_3]\begin{bmatrix} 1 - \gamma & 0 \\ 0 & I - \gamma\Lambda \end{bmatrix}\Omega = \Omega^\top\begin{bmatrix} 1 - \gamma & 0 \\ 0 & I - \gamma\Lambda \end{bmatrix}\Omega$. Hence, $(\Phi^\top L\Phi)^{-1} = \begin{bmatrix} (1 - \gamma)^{-1} & 0 \\ 0 & (u^\top(I - \gamma\Lambda)u)^{-1} \end{bmatrix}$ with $u = (w, (1 - w))^\top$. Note that $u^\top(I - \gamma\Lambda)u = \omega^2\lambda_{1,1} + (1 - \omega)^2\lambda_{1,1}$

The TD value function is given by

$$\Phi(\Phi^\top L\Phi)^{-1}\Phi^\top = [e_1, e_2, e_3]\Omega\begin{bmatrix} (1 - \gamma)^{-1} & 0 \\ 0 & (u^\top(I - \gamma\Lambda)u)^{-1} \end{bmatrix}\Omega^\top[e_1, e_2, e_3]^\top$$
$$= [e_1, e_2, e_3]\begin{bmatrix} (1 - \gamma)^{-1} & 0 \\ 0 & u(u^\top(I - \gamma\Lambda)u)^{-1}u^\top \end{bmatrix}[e_1, e_2, e_3]^\top$$
$$= (1/(1 - \gamma)e_1 e_1^\top + \omega^2 e_2 e_2^\top + \omega(1 - \omega)e_3 e_2^\top + \omega(1 - \omega)e_2 e_3^\top + (1 - \omega)^2 e_3 e_3^\top)/(\omega^2\lambda_{1,1} + (1 - \omega)^2\lambda_{1,1})$$

Now $\|\Phi(\Phi^\top L\Phi)^{-1}\Phi^\top - V^\pi\|_F^2$ is independent of $\omega$. $\qquad\square$

We now proceed to the proof of Proposition 3. Before that, we introduce some necessary notations and lemmas.

## D.1. Notations

Let $O(S, d) := \{A \in \mathbb{R}^{S \times d} : A^\top A = I\}$.

**Definition 2.** *Let $A, B \in O(S, d)$. The principle angles $\Theta$ between $A$ and $B$ are given by writing the SVD of $A^\top B = U \cos\Theta V^\top$.*

**Definition 3.** *Let $A, B \in O(S, d)$ with principle angles $\Theta$. We define the distance $d(A, B)$ as $d(A, B) := \|\sin\Theta\|_{\mathrm{op}}$.*

**Proposition 4.** *Let $A, B \in O(S, d)$. We have the following identities:*

$$d(A, B) = \|AA^\mathsf{T} - BB^\mathsf{T}\|_{\mathrm{op}} = \|\sin\Theta\|_{\mathrm{op}} = \|A^\mathsf{T}\bar{B}\|_{\mathrm{op}},$$

*where $\bar{B} \in O(S, S - d)$ satisfies $BB^\mathsf{T} + \bar{B}\bar{B}^\mathsf{T} = I$.*

### D.2. Approximate matrix decompositions

**Lemma 9** (Deterministic error bound). *Let $A$ be an $S \times S$ matrix. Fix $d \leqslant S$, and partition the SVD of $A$ as:*

$$A = \begin{bmatrix} U_1 & U_2 \end{bmatrix} \begin{bmatrix} \Sigma_1 & 0 \\ 0 & \Sigma_2 \end{bmatrix} \begin{bmatrix} V_1^\mathsf{T} \\ V_2^\mathsf{T} \end{bmatrix},$$

*where $\Sigma_1$ is $d \times d$ (the dimensions of all the other factors are determined by this selection). Put $A_d := U_1\Sigma_1 V_1^\mathsf{T}$ as the rank-d approximation of $A$. Let $\Omega$ be an $S \times \ell$ test matrix ($\ell \geqslant d$). Put $Y = A\Omega$, $\Omega_1 = V_1^\mathsf{T}\Omega$ and $\Omega_2 = V_2^\mathsf{T}\Omega$. We have that:*

$$\|(I - P_Y)A_k\|_{\mathrm{op}}^2 \leqslant \|\Sigma_2\Omega_2\Omega_1^\dagger\|_{\mathrm{op}}^2.$$

*Proof.* This proof is adapted from Theorem 9.1 of Halko et al. (2011).

Write $A_d = \hat{U}\hat{\Sigma}\hat{V}^\mathsf{T}$ the full SVD of $A_d$. By invariance of the spectral norm to unitary transformations,

$$\|(I - P_Y)A_d\|_{\mathrm{op}}^2 = \|\hat{U}^\mathsf{T}(I - P_Y)\hat{U}(\hat{U}^\mathsf{T}A_d)\|_{\mathrm{op}}^2 = \|(I - P_{\hat{U}^\mathsf{T}Y})(\hat{U}^\mathsf{T}A_d)\|_{\mathrm{op}}^2$$

Assume the diagonal entries of $\Sigma_2$ are not all strictly positive. Then $\Sigma_2$ is zero as a consequence of the ordering of the singular values.

$$\mathrm{range}(\hat{U}^\mathsf{T}Y) = \mathrm{range}\begin{bmatrix} \Sigma_1\Omega_1 \\ 0 \end{bmatrix} = \mathrm{range}\begin{bmatrix} \Sigma_1 V_1^\mathsf{T} \\ 0 \end{bmatrix} = \mathrm{range}(\hat{U}^\mathsf{T}A_d)$$

So we can conclude that $\|(I - P_Y)A_d\|_{\mathrm{op}}^2 = 0$ assuming that $V_1^\mathsf{T}$ and $\Omega_1$ have full row rank.

Now assume that the diagonal entries of $\Sigma_1$ are strictly positive. Let $Z = \hat{U}^\mathsf{T}Y \cdot \Omega_1^\dagger\Sigma_1^{-1} = \begin{bmatrix} I_d \\ F \end{bmatrix}$ with $F = \Sigma_2\Omega_2\Omega_1^\dagger\Sigma_1^{-1} \in \mathbb{R}^{(S-d)\times d}$.

By construction, $\mathrm{range}(Z) \subset \mathrm{range}(\hat{U}^\mathsf{T}Y)$, hence we have,

$$\|(I - P_{\hat{U}^\mathsf{T}Y})(\hat{U}^\mathsf{T}A_d)\|_{\mathrm{op}}^2 \leqslant \|(I - P_Z)\hat{U}^\mathsf{T}A_d\|_{\mathrm{op}}^2 \leqslant \|A_d^\mathsf{T}\hat{U}(I - P_Z)\hat{U}^\mathsf{T}A_d\|_{\mathrm{op}} \leqslant \|\hat{\Sigma}(I - P_Z)\hat{\Sigma}\|_{\mathrm{op}}$$

Following the proof from Theorem 9.1 of Halko et al. (2011), we have

$$(I - P_Z) \preccurlyeq \begin{bmatrix} F^\mathsf{T}F & B \\ B^\mathsf{T} & I_{S-d} \end{bmatrix}$$

where $B = -(I_d - F^\mathsf{T}F)^{-1}F^\mathsf{T} \in \mathbb{R}^{d\times(S-d)}$.

Consequently, we have

$$\hat{\Sigma}(I - P_Z)\hat{\Sigma} \preccurlyeq \begin{bmatrix} \Sigma_1 F^\mathsf{T}F\Sigma_1 & 0 \\ 0 & 0 \end{bmatrix}$$

$\hat{\Sigma}(I - P_Z)\hat{\Sigma}$ is PSD by the conjugation rule, hence the matrix on the right hand side is PSD too. It follows that

$$\|\hat{\Sigma}(I - P_Z)\hat{\Sigma}\|_{\mathrm{op}} \leqslant \|\Sigma_1 F^\mathsf{T}F\Sigma_1\|_{\mathrm{op}} = \|F\Sigma_1\|_{\mathrm{op}}^2 = \|\Sigma_2\Omega_2\Omega_1^\dagger\|_{\mathrm{op}}^2$$

$\square$

**Lemma 10** (Average spectral error). *Let $A$ be an $S \times S$ matrix with singular values $\sigma_1 \geqslant \sigma_2 \geqslant \ldots$. Fix a target rank $2 \leqslant d \leqslant S$ and an oversampling parameter $p \geqslant 2$ where $p + d \geqslant S$. Draw and $S \times (d + p)$ standard gaussian matrix $\Omega$ and construct the sample matrix $Y = A\Omega$. Then, we have*

$$\mathbb{E}\|(I - P_Y)A_d\|_{\mathrm{op}} \leqslant \sqrt{\frac{d}{p-1}}\sigma_{d+1} + \frac{e\sqrt{d+p}}{p}\left(\sum_{j=d+1}^{S} \sigma_j^2\right)^{1/2}.$$

*Proof.* By Lemma 9 and linearity of the expectation, we have

$$\mathbb{E}\|(I - P_Y)A_d\|_{\mathrm{op}} \leqslant \mathbb{E}\|\Sigma_2\Omega_2\Omega_1^\dagger\|_{\mathrm{op}}$$

$$\leqslant \sqrt{\frac{d}{p-1}}\sigma_{d+1} + \frac{e\sqrt{d+p}}{p}\left(\sum_{j=d+1}^{S} \sigma_j^2\right)^{1/2},$$

where the last inequality comes from Theorem 10.6 of Halko et al. (2011). $\square$

**Lemma 11.** *Let $A \in \mathbb{R}^{m \times n}$, and fix a $d < n$. Let $\sigma_1 \geqslant \sigma_2 \geqslant \ldots \geqslant \sigma_n$ denote the singular values of $M$ listed in decreasing order, and suppose that $\sigma_k > 0$. Let $A_d$ denote the rank-d approximation of $A$. Fix any matrix $Y \in \mathbb{R}^{m \times T}$. We have:*

$$\|(I - P_Y)A_k\|_{\mathrm{op}} \geqslant \|(I - P_Y)P_{A_k}\|_{\mathrm{op}}\sigma_k.$$

*Proof.* Decompose $P_Y^\perp A_k$ as:
$$P_Y^\perp A_k = P_Y^\perp P_{A_k} A_k$$

$$\|P_Y^\perp A_k\|_{\mathrm{op}} = \|P_Y^\perp P_{A_k} A_k\|_{\mathrm{op}} \geqslant \|P_Y^\perp P_{A_k}\|_{\mathrm{op}}\|A_k\|_{\mathrm{op}} = \|P_Y^\perp P_{A_k}\|_{\mathrm{op}}\sigma_k$$

where the inequality comes from the sub-multiplicativity of the the operator norm $\square$

**Proposition 5.** *Let $A$ be an $S \times S$ matrix with singular values $\sigma_1 \geqslant \sigma_2 \geqslant \ldots$. Fix a target rank $2 \leqslant d \leqslant n$ and an oversampling parameter $p \geqslant 2$ where $p + d \geqslant S$. Draw and $n \times (d + p)$ standard gaussian matrix $\Omega$ and construct the sample matrix $Y = A\Omega$. Then, we have*

$$\mathbb{E}\|(I - P_Y)P_{A_d}\|_{\mathrm{op}} \leqslant \sqrt{\frac{d}{p-1}}\frac{\sigma_{d+1}}{\sigma_d} + \frac{e\sqrt{d+p}}{p}\left(\sum_{j=d+1}^{S} \frac{\sigma_j^2}{\sigma_d^2}\right)^{1/2}.$$

*Proof.* By Lemma 11 and linearity of the expectation, we have

$$\frac{1}{\sigma_d}\mathbb{E}\|(I - P_Y)A_d\|_{\mathrm{op}} \geqslant \mathbb{E}\|(I - P_Y)P_{A_d}\|_{\mathrm{op}}$$

Now applying Lemma 10, we have

$$\sqrt{\frac{d}{p-1}}\frac{\sigma_{d+1}}{\sigma_d} + \frac{e\sqrt{d+p}}{p}\left(\sum_{j=d+1}^{S} \frac{\sigma_j^2}{\sigma_d^2}\right)^{1/2} \geqslant \mathbb{E}\|(I - P_Y)P_{A_d}\|_{\mathrm{op}}$$

$\square$

Observe that, as the oversampling factor $p$ grows, the RHS tends to zero. However, the dependence will be something like $p \gtrsim 1/\varepsilon^2$, if you want the RHS to be $\leqslant \varepsilon$. This actually makes sense I think– you are using concentration of measure to increase the accuracy, so you should pay $1/\varepsilon^2$ sample complexity.

### D.3. Analysis

**Proposition 3** (MC Error bound). *Let $G \in \mathbb{R}^{S \times T}$ be a sample from a standard gaussian distribution and assume $d \leqslant T$. Let $F_d$ be the top-d left singular vectors of the successor representation $(I - \gamma P^\pi)^{-1}$ and $\hat{F}_d$ be the top left singular vectors of $(I - \gamma P^\pi)^{-1}G$. Denote $\sigma_1 \geqslant \sigma_2 \geqslant ... \geqslant \sigma_S$ the singular values of the SR and $\mathrm{dist}(F_d, \hat{F}_d)$ the $\sin \theta$ distance between the subspaces spanned by $F_d$ and $\hat{F}_d$. We have*

$$\mathbb{E}[\mathrm{dist}(F_d, \hat{F}_d)] \leqslant \sqrt{\frac{d}{T - d - 1}} \frac{\sigma_{d+1}}{\sigma_d} + \frac{e\sqrt{T}}{T - d} \left( \sum_{j=d+1}^{n} \frac{\sigma_j^2}{\sigma_d^2} \right)^{\frac{1}{2}}$$

*Proof.* Let $l \in \{d,...,S\}$. $F_l \in O(S, l)$ be the top $l$ left singular vectors of $(I - \gamma P^\pi)^{-1}$ and $\hat{F}_l \in O(S, d)$ be the top left singular vectors of $(I - \gamma P^\pi)^{-1}G$.

$$\begin{aligned}
d(F_d, \hat{F}_d) &= \|\hat{F}_d^\top F_d^\perp\|_{\mathrm{op}} \\
&= \|P_{\hat{F}_d} P_{F_d}^\perp\|_{\mathrm{op}} \\
&\leqslant \|P_{L^{-1}G} P_{F_d}^\perp\|_{\mathrm{op}} \text{ as } \mathrm{span}(\hat{F}_d) \subseteq \mathrm{span}(L^{-1}G) \\
&= \|\hat{F}_T^\top F_d^\perp\|_{\mathrm{op}} \\
&= \|F_d^\top \hat{F}_T^\perp\|_{\mathrm{op}} \\
&= \|P_{F_d} P_{\hat{F}_T}^\perp\|_{\mathrm{op}} \\
&= \|P_{\hat{F}_T}^\perp P_{F_d}\|_{\mathrm{op}} \text{ by symmetry of the projection matrices} \\
&= \|(I - P_{\hat{F}_T}) P_{F_d}\|_{\mathrm{op}} \\
&= \|(I - P_{L^{-1}G}) P_{(L^{-1})_d}\|_{\mathrm{op}} \\
&\leqslant \frac{1}{\sigma_d} \|(I - P_{L^{-1}G})(L^{-1})_d\|_{\mathrm{op}} \text{ by Lemma 11}
\end{aligned}$$

Now taking the expectation with respect to $G$ and applying Proposition 5,

$$\mathbb{E}[d(F_d, \hat{F}_d)] \leqslant \sqrt{\frac{d}{T - d - 1}} \frac{\sigma_{d+1}}{\sigma_d} + \frac{e\sqrt{T}}{T - d} \left( \sum_{j=d+1}^{n} \frac{\sigma_j^2}{\sigma_d^2} \right)^{1/2}.$$

$\square$