# OpenReview forum: "Bootstrapped Representations in Reinforcement Learning"
_ICLR.cc/2023/Workshop/RRL — RRL 2023 Poster_

### Official Review · Reviewer_i3xF · 2023-02-26
**interesting analysis, could be better explained/developed**

**Rating:** 2
**Confidence:** 4

**Review:**

The research investigates the application of pre-training to enhance the capability of deep reinforcement learning agents to handle vast or continuous state spaces. The authors suggest new unsupervised pre-training guidelines and contrast the features gained using various pre-training techniques. The authors demonstrate how these pre-trained representations are successful at accelerating online learning across a variety of fields.

Comments:
Are these works related? [1,2]

Strengths:

1. The paper offers an interesting theoretical analysis.
2. The study suggests new unsupervised pre-training procedures.
3. Empirical success on some simple datasets.

Weaknesses:

1. The theoretical research is restricted to an idealized scenario, and it is uncertain how much the findings apply to more complicated systems.
2. The empirical evaluation is limited to two domains: four-room (Sutton et al., 1999) and Mountain Car (Moore, 1990).
3. The paper listed a bunch of theoretical results but did not provide enough insight/comparison.
4. The proposed pre-training methods are not compared to other SOTA pre-training strategies in the research.
5. Limitations are not thoroughly discussed in the paper.

[1] Eysenbach, Benjamin, et al. "Contrastive learning as goal-conditioned reinforcement learning." NeurIPS 2022.
[2] Eysenbach, Benjamin, et al. "Imitating past successes can be very suboptimal." NeurIPS 2022.

---

### Official Review · Reviewer_t2La · 2023-03-02
**Interesting study of representation learning for value based deep RL**

**Rating:** 3
**Confidence:** 1

**Review:**

Authors study the problem of representation learning in Deep RL where they look at representations are affected by
i) choice of learning method such as TD learning
ii) also the way in which they are trained (i.e stop gradient)
They discuss conditions where TD and Monte-carlo representations are same/different as per the cumulant matrix G.

To analyse which approach leads to better representation, they learn the representation and then perform policy evaluation using this representation. They then study how representations are dependent on the auxiiary task characterized through cumulant matrix. As a particular case, they study random cumulants, which is of practical importance.

Overall the paper is well-written, the concepts are explained in a good way and they dive deep into this problem with interesting potential directions. My expertise in this area is limited to give a more refined judgement.